# Facile conversion of water to functional molecules and cross-linked polymeric films with efficient clusteroluminescence

Bo Song[1,5], Jianyu Zhang [1,5], Jiadong Zhou [2], Anjun Qin [2,3], Jacky W. Y. Lam [1] ✉ & Ben Zhong Tang [1,3,4] ✉

Exploring approaches to utilize abundant water to synthesize functional molecules and polymers with efficient clusteroluminescence properties is highly significant but has yet to be reported. Herein, a chemistry of water and alkyne is developed. The synthesized products are proven as nonaromatic clusteroluminogens that could emit visible light. Their emission colors and luminescent efficiency could be adjusted by manipulating through-space interaction using different starting materials. Besides, the free-standing polymeric films with much high photoluminescence quantum yields (up to 45.7%) are in situ generated via a water-involved interfacial polymerization. The interfacial polymerization-enhanced emission of the polymeric films is observed, where the emission red-shifts and efficiency increases when the polymerization time is prolonged. The synthesized polymeric film is also verified as a Janus film. It exhibits a vapor-triggered reversible mechanical response which could be applied as a smart actuator. Thus, this work develops a method to synthesize clusteroluminogens using water, builds a clear structure-property relationship of clusteroluminogens, and provides a strategy to in situ construct functional water-based polymeric films.

Water ($H_2O$), occupying 71% of our earth's surface area, is life's origin. Nature uses $H_2O$ as a feedstock to synthesize various complex species continuously. For example, chloroplasts can convert $H_2O$ and carbon dioxide into carbohydrates and oxygen through photosynthesis, which is one of nature's most critical $H_2O$-involved reactions. Most species prepared using water as a reactant are nonaromatic and non-conjugated, serving as commodities for everyday use. They are hardly considered specialty materials, especially in the field of optoelectronics. According to traditional molecular photophysical theories based on through-bond conjugation (TBC), excellent luminogens always contain largely conjugated units such as phenyl, carbazole, and fluorene[1–7]. Hence, researchers take it for granted that the $H_2O$-based species exhibit no property for optoelectronic applications.

However, in recent years, some nonaromatic or nonconjugated small molecules and polymers which could emit visible light in the aggregate state have sprung up, breaking people's perception of luminescence. This non-conventional phenomenon is termed cluster-oluminescence (CL), and the luminophores with CL properties are named clusteroluminogens[8–31]. However, most clusteroluminogens only emit light within the blue-light region with low photoluminescence

[1]Department of Chemistry, Hong Kong Branch of Chinese National Engineering Research Center for Tissue Restoration and Reconstruction, and Guangdong-Hong Kong-Macau Joint Laboratory of Optoelectronic and Magnetic Functional Materials, The Hong Kong University of Science and Technology, Clear Water Bay, 999077 Kowloon, Hong Kong, China. [2]State Key Laboratory of Luminescent Materials and Devices, Guangdong Provincial Key Laboratory of Lumines-cence from Molecular Aggregates, South China University of Technology, 510640 Guangzhou, China. [3]Center for Aggregation-Induced Emission, AIE Institute, South China University of Technology, 510640 Guangzhou, China. [4]Shenzhen Institute of Aggregate Science and Technology, School of Science and Engineering, The Chinese University of Hong Kong, 518172 Shenzhen, Guangdong, China. [5]These authors contributed equally: Bo Song, Jianyu Zhang. ✉e-mail: chjacky@ust.hk; tangbenz@cuhk.edu.cn

quantum yield (PLQY). Moreover, the mechanism behind CL is unclear due to the ambiguous structure-property relationship. Among reported works, most clusteroluminogens have electron-rich heteroatoms (e.g., O, N, S, etc.) in their structures with abundant lone-pair (*n*) electrons. Those electron-rich atoms are significant for the formation of wide-spread through-space interactions (TSI) (e.g., *n*···*n*, *n*···*π*, and *π*···*π* interactions), which might be the main reason for visible light from clusteroluminogens[32,33]. However, TSI-based theory is not established, and manipulating TSI to adjust the photophysical properties of clusteroluminogens is still a challenge[34,35].

Learning from nature and exploring approaches for converting water to functional molecules and polymers with CL properties is highly significant but has yet to be reported. Unlike the complex processes of photosynthesis in chloroplasts, the $H_2O$-involved reaction should be simple and robust. It would be better if the resultant nonaromatic products had well-defined structures and could serve as a versatile platform for TSI manipulation, mechanistic study, and advanced applications.

Keeping this idea in mind, we reported a chemistry of $H_2O$ and alkyne in this work. This reaction could happen under air at room temperature in the presence of 1,4-diazabicyclo[2.2.2]octane (DABCO) as an organic catalyst. The resultant nonaromatic products are verified as clusteroluminogens that could emit visible light in the aggregate state. Tunable emission colors and luminescent efficiency of these

clusteroluminogens could be further manipulated using different starting materials. A clear structure-property relationship was built for the mechanistic study of TSI.

Polymerization-induced emission (PIE) as a concept indicates that polymerization could effectively improve CL behavior[12,15,16,36–43]. Only a few PIE-related works have been reported till now. Much more investigations need to be done to investigate the mechanism behind PIE. Herein, the cross-linked polymers were synthesized via an $H_2O$-involved interfacial polymerization. Nonaromatic polymeric films with much higher PLQY (up to 45.7%) could be in situ generated, and the interfacial polymerization-enhanced emission phenomenon was observed. The obtained thin film is a Janus film because the two sides show different roughness and vapor absorption capacities. Thus, it also exhibits a vapor-triggered reversible mechanical response which could be applied as a vapor-responsive actuator.

## Results

### $H_2O$-involved reaction for CLgen synthesis

A reaction of methyl propiolate (MP) and $H_2O$ was developed at room temperature under air in the presence of DABCO (Fig. 1a). Two products, *EE*-DMODA and *EZ*-DMODA, were isolated and purified via a silica gel column (molar ratio: 24:1). The structures of the two products were characterized by nuclear magnetic resonance (NMR) spectra and high-resolution mass spectra (HRMS) (Supplementary Fig. 1). For the

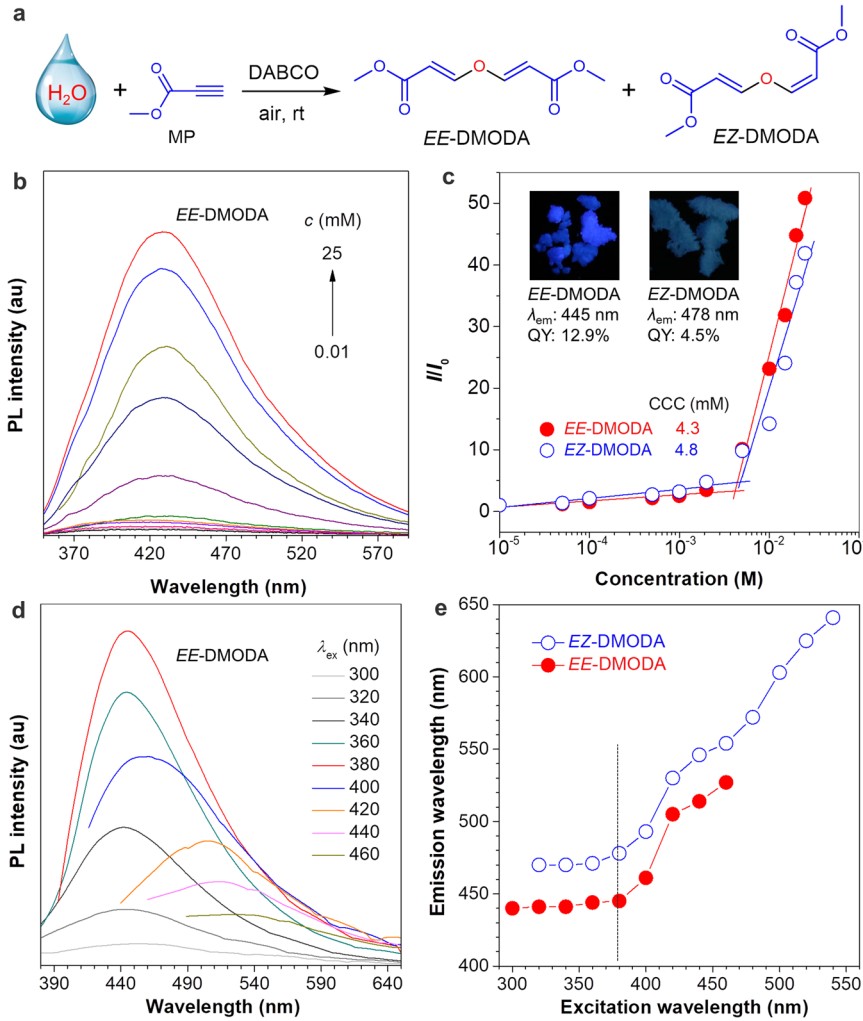

**Fig. 1 | Nonaromatic clusteroluminogens from water. a** One-step conversion from $H_2O$ and MP to nonaromatic clusteroluminogens. **b** PL spectra of *EE*-DMODA in THF with different concentrations ($\lambda_{ex}$ = 330 nm). **c** Plots of relative PL intensity ($I/I_0$) versus concentration, $I_0$ = PL intensity at $10^{-5}$ M. Insets: photophysical

properties and photographs of solid powders taken under UV light. **d** PL spectra of *EE*-DMODA in the solid state. **e** Plots of emission wavelengths versus excitation wavelength.

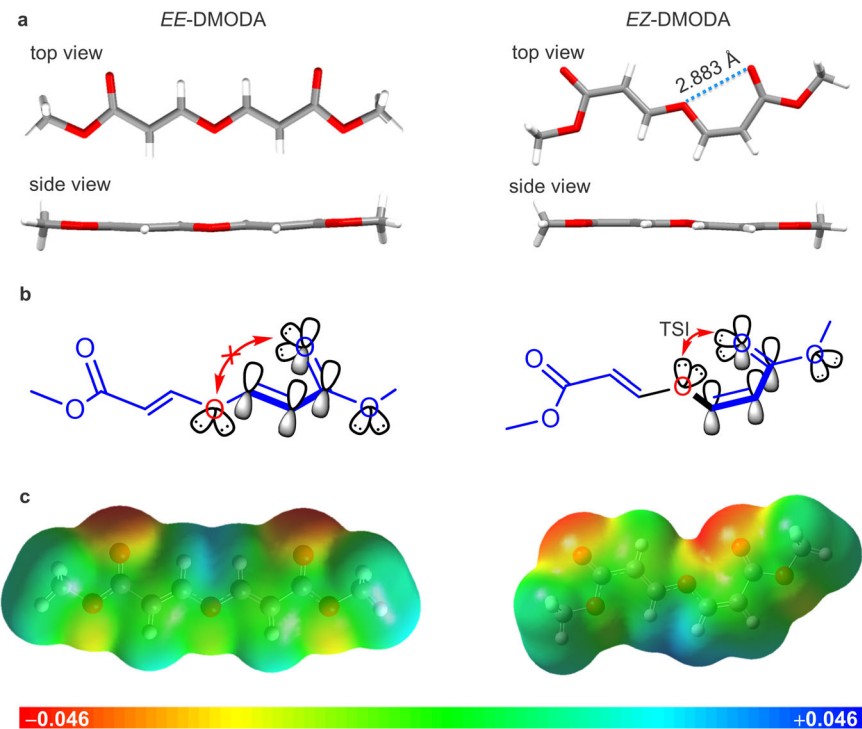

**Fig. 2 | Single-molecule analysis. a, b** Single-crystal structures and orbital diagrams of *EE*-DMODA and *EZ*-DMODA. **c** Electronic static potential mapped on the isosurface of electronic density. A negative electrostatic potential (red) represents a high electron-density region, while the positive one (blue) corresponds to a low electron-density region. The values of the potential are given in au.

[1]H NMR spectrum of *EE*-DMODA, two typical peaks at 7.58 and 5.66 ppm are assigned to the resonances of the HC = CH groups. The coupling constant of the HC = CH group is 12.0 Hz, confirming the *E*-isomer of the two double bonds in *EE*-DMODA. Regarding the [1]H NMR spectrum of *EZ*-DMODA, four peaks at 7.57, 6.70, 5.71, and 5.21 ppm are assigned to the resonances of two types of HC = CH groups. Their coupling constants of 12.4 and 7.2 Hz correspond to the *E*-isomer and *Z*-isomer of the two double bonds in *EZ*-DMODA, respectively.

After confirming their structures, the photophysical properties of *EE*-DMODA and *EZ*-DMODA were studied. Their UV-Vis absorption and photoluminescent (PL) spectra in dilute solution were measured (Fig. 1b, Supplementary Figs. 2 and 3). *EE*-DMODA and *EZ*-DMODA exhibit almost the same absorption band at around 245 nm and are nonemissive in dilute tetrahydrofuran (THF) solution ($10^{-5}$M). However, with the increased concentration, their PL intensities increase slightly at low concentrations but then enhance sharply at high concentrations, which is a typical characteristic of clusteroluminogens[8,11,21]. It is hard for molecules to interact with each other to form clusters at low concentrations, so the PL intensity shows no apparent change. However, when the concentration is increased to a critical point, defined as critical cluster concentration (CCC), the clusters are formed via sufficient intermolecular interactions, leading to a remarkable enhancement in PL intensity. Accordingly, the CCC values of *EE*-DMODA and *EZ*-DMODA were determined as 4.3 and 4.8 mM, respectively (Fig. 1c). Then their photophysical properties in the solid state were also tested (Fig. 1d, Supplementary Figs. 4–7 and Supplementary Table 1). They show unmatched absorption and excitation spectra and excitation-dependent luminescence as most reported works[8,11,12,21,32,33]. *EZ*-DMODA always shows a longer emission wavelength than *EE*-DMODA upon the same excitation wavelength ($\lambda_{ex}$). However, *EE*-DMODA owns an absolute PLQY of 12.9 %, which is higher than *EZ*-DMODA at $\lambda_{ex}$ of 380 nm.

Their single-crystal structures were determined to gain deeper insights into the photophysical properties of these nonaromatic clusteroluminogens (Supplementary Table 2). They both exhibit almost planar conformation (Fig. 2a). Significantly, an intramolecular O···O interaction with a distance of 2.883 Å in *EZ*-DMODA is observed, which is shorter than the sum of the van der Waals radius of two oxygen atoms (3.04 Å)[32]. Thus, there is a typical *n*···*n* TSI in *EZ*-DMODA (Fig. 2b). Further, the excited-state electronic static potential based on electronic density was mapped (Fig. 2c). *EE*-DMODA exhibits low electronic density at the central oxygen atom. However, the same oxygen atom in *EZ*-DMODA shows a higher electronic density due to the adjacent carbonyl oxygen, further supporting the intramolecular TSI in *EZ*-DMODA. This intramolecular interaction also explains the longer-wavelength emission of *EZ*-DMODA compared with *EE*-DMODA. Frontier molecular orbitals of *EE*-DMODA and *EZ*-DMODA were also calculated (Supplementary Fig. 8). The result indicates that *EZ*-DMODA processes a narrower energy gap (4.427 eV) between the highest occupied molecular orbital (HOMO) and the lowest unoccupied molecular orbital (LUMO) than that of *EE*-DMODA (4.507 eV), which was consistent with the experimental results.

Then the intermolecular interactions of *EE*-DMODA and *EZ*-DMODA were analyzed. As illustrated in Fig. 3, there are multiple hydrogen bonds and *n*···*π* interactions in the networks of *EE*-DMODA and *EZ*-DMODA. Such three-dimensional intermolecular interactions are the main reasons for the formation of clusters, further achieving CL with visible light. Especially for dimer 1 and dimer 2 of *EE*-DMODA, the C = O···C = C and O···C = C distances are 3.228–3.591 Å, which are shorter than those in dimer 3 and dimer 4 of *EZ*-DMODA (3.394–3.619 Å). Thus, *EE*-DMODA shows stronger intermolecular TSI than *EZ*-DMODA, resulting in higher PLQY of solid-state *EE*-DMODA.

## Manipulation of TSI

Thanks to the versatile $H_2O$-involved chemistry, different starting materials could be utilized to synthesize clusteroluminogens and realize the manipulation of TSI. Hence, different water, such as heavy oxygen water ($H_2^{18}O$) and heavy water ($D_2O$), was used as the starting

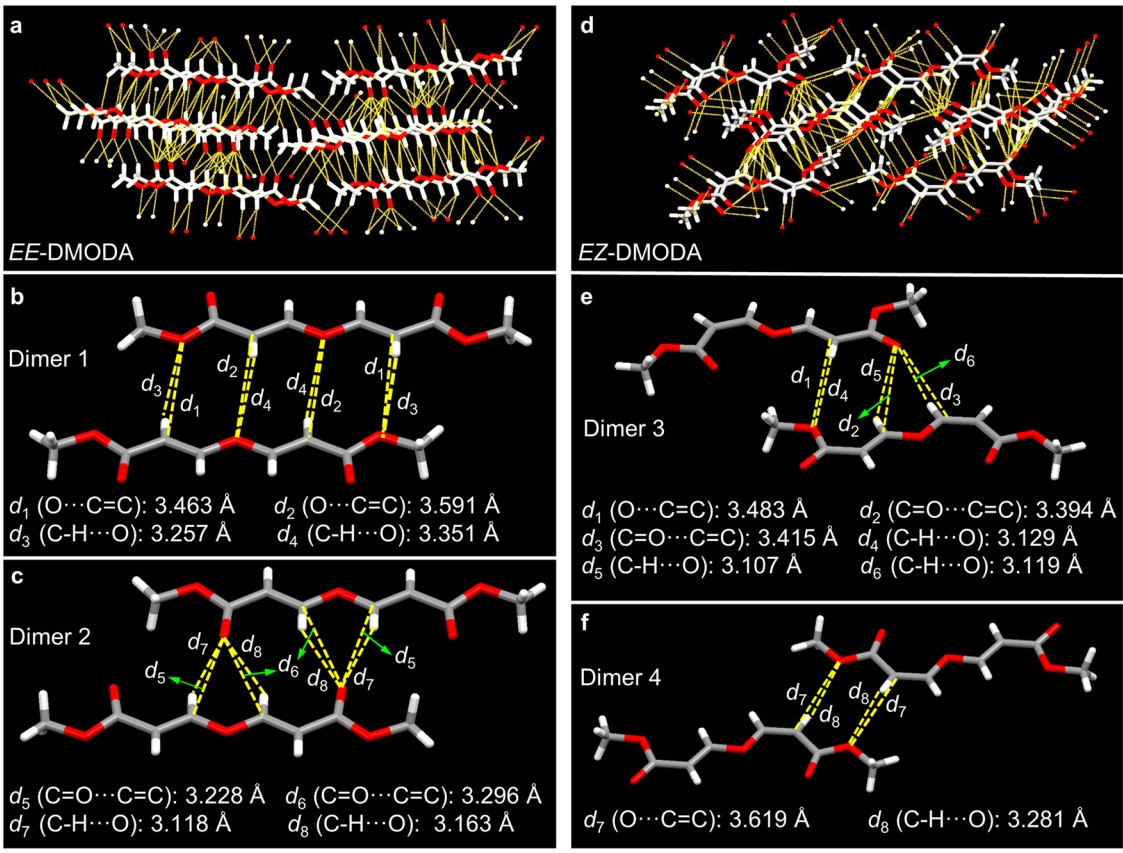

**Fig. 3 | Intermolecular interactions analysis. a, d** Global hydrogen bonds, and *n*···*π* interactions of *EE*-DMODA and *EZ*-DMODA, respectively. The C = O···C = C, O···C = C, and C-H···O interactions in different dimers from **b, c** *EE*-DMODA and **e, f** *EZ*-DMODA, respectively.

material to obtain different clusteroluminogens (Fig. 4). Luckily, the isotope effect of clusteroluminogens is also investigated. $H_2^{18}O$ was used to react with MP to obtain *EE*-DMODA-$^{18}$O and *EZ*-DMODA-$^{18}$O. The oxygen atoms of the ether group in these two compounds were proved to come from $H_2^{18}O$ via HRMS (Supplementary Fig. 9). Further, their photophysical properties were tested and compared (Supplementary Figs. 10–21). Notably, the heavy oxygen-substituted clusteroluminogens show red-shifted emission and higher PLQY than the corresponding normal compounds of *EE*-DMODA and *EZ*-DMODA. Luckily, the single-crystal structure of *EE*-DMODA-$^{18}$O was obtained to figure out the reason (Supplementary Table 2). *EE*-DMODA-$^{18}$O displays almost identical molecular conformation, packing mode, and TSI as *EE*-DMODA (Fig. 4b, c). However, for dimer 1′ and dimer 2′ of *EE*-DMODA-$^{18}$O, intermolecular C = O···C = C and O···C = C distances are shorter than those in dimer 1 and dimer 2 of *EE*-DMODA (Fig. 3). The above results suggest that the heavy oxygen-substituted clusteroluminogens possess stronger TSI, which benefits their luminescent behaviors.

Then the reaction of $D_2O$ and MP was carried out. As shown in Fig. 4d, because a very large excess of $D_2O$ was used in this reaction and the ethynyl proton of MP is highly active, it could exchange with D in $D_2O$. Next, the deuterated MP reacted with $D_2O$ to generate *EE*-DMODA-$D_4$ and *EZ*-DMODA-$D_4$, respectively. In their $^1$H NMR spectra (Supplementary Fig. 22), some small peaks could also be found, which are assigned to the resonances of HC = CH groups due to incomplete deuteration. The deuterated efficiency was deduced to be 91 atom% according to the integration of different peaks in their $^1$H NMR spectra. Then their photophysical properties were also tested (Supplementary Figs. 10–21). Unlike heavy oxygen-substituted clusteroluminogens, the deuterated clusteroluminogens (*EE*-DMODA-$D_4$ and *EZ*-DMODA-$D_4$) show blue-shifted emission and lower PLQY than the corresponding

normal compounds. Due to the incomplete deuterated efficiency, many different compounds with irregular structures were obtained (Supplementary Fig. 23), which made their crystallization difficult. Powder X-ray diffraction (PXRD) patterns also suggest that the deuterated clusteroluminogens show poorer crystallinity than the normal and heavy oxygen-substituted clusteroluminogens (Supplementary Figs. 24 and 25). Their irregular structures weaken intermolecular interactions, resulting in weak and hypsochromic emission of deuterated clusteroluminogens.

Notably, the isotope effect of traditional luminogens always results in enhanced PLQY but unchanged emission wavelength[44–49], which is different from that of clusteroluminogens. According to our study, the isotope effect of clusteroluminogens could change both PLQY and emission wavelength. The reason is that the CL property does not merely depend on a single molecule but is closely related to intermolecular and intramolecular TSI. The isotope effect could significantly influence the TSI of clusteroluminogens, further affecting the photophysical behaviors of clusteroluminogens.

As an extension, a different alkyne MEK could be utilized to react with $H_2O$. Unlike the above reactions, only the *E*-isomer compound of *EE*-OBBO was generated, as shown in Fig. 5a. The structure of *EE*-OBBO was confirmed via NMR spectra and HRMS, and its photophysical properties were also measured (Supplementary Figs. 26–32). *EE*-OBBO also shows typical characteristics of clusteroluminogens, such as concentration-dependent emission, unmatched absorption and excitation spectra, and excitation-dependent luminescence. The dynamic light scattering (DLS) measurement was used to track the evolution of the cluster. The large size of clusters could be detected by DLS when the concentration of EE-OBBO exceeded CCC. However, no signal was detected when the concentration was lower than the CCC point. Moreover, it was observed that the size of the clusters increased when

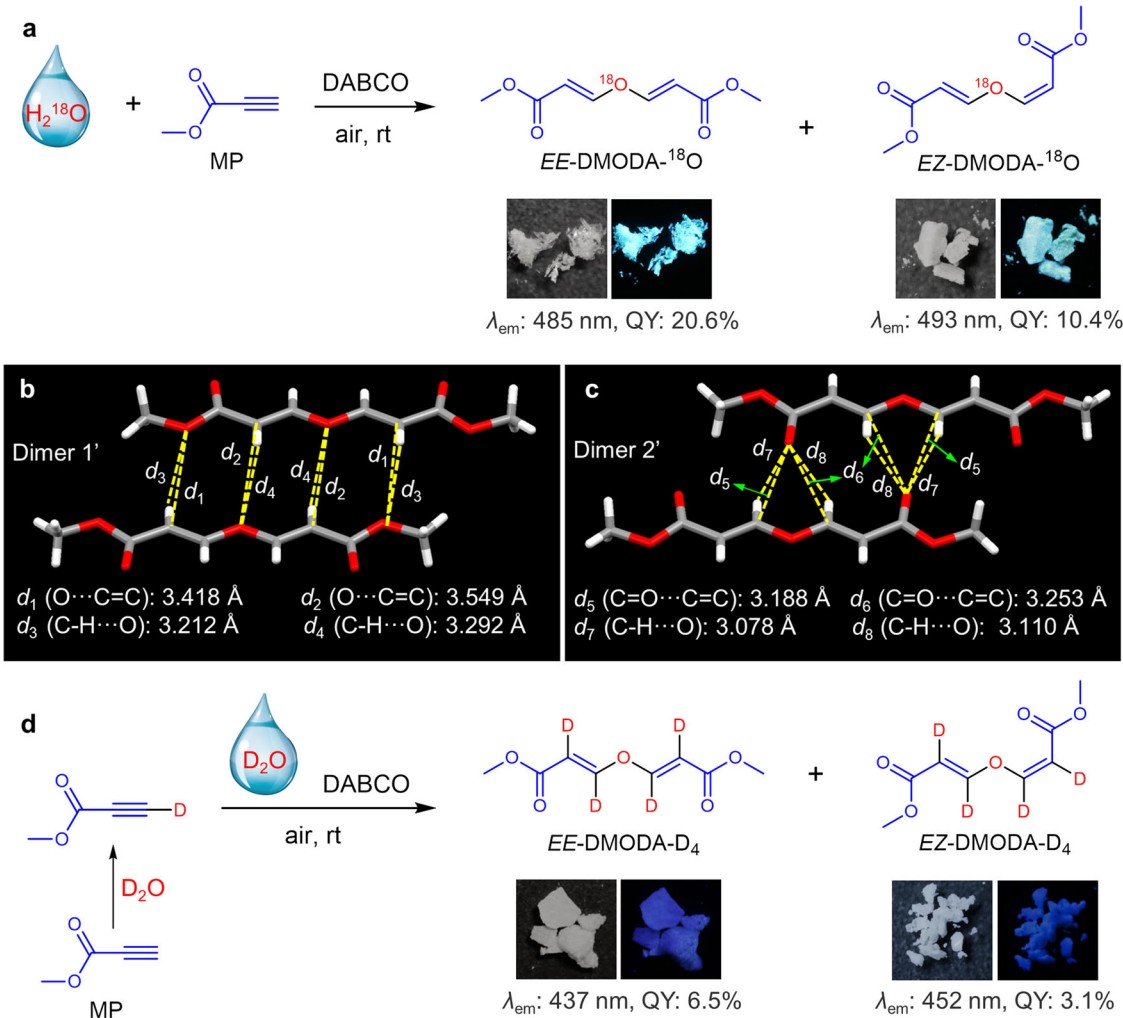

**Fig. 4 | Isotope effect for clusteroluminogens. a** One-step conversion from $H_2^{18}O$ and MP to nonaromatic clusteroluminogens of *EE*-DMODA-$^{18}$O and *EZ*-DMODA-$^{18}$O. Photophysical properties and photographs of solid powders taken under daylight (left) and UV light (right) are given. **b**, **c** The $C=O\cdots C=C$, $O\cdots C=C$, and $C-H\cdots O$ interactions in different dimers from *EE*-DMODA-$^{18}$O. **d** One-step conversion from $D_2O$ and MP to nonaromatic clusteroluminogens of *EE*-DMODA-$D_4$ and *EZ*-DMODA-$D_4$. Photophysical properties and photographs of solid powders taken under daylight (left) and UV light (right) are given.

the concentration increased from 10 to 25 mM, resulting in red-shifted emission and enhanced intensity (Supplementary Figs. 33–36). Besides, it shows red-shifted emission and higher PLQY in the solid state than *EE*-DMODA. Two types of dimers are observed in the crystal packing of *EE*-OBBO. The $C=O\cdots C=C$ and $O\cdots C=C$ distances are 3.173–3.559 Å, which are shorter than those of *EE*-DMODA (Fig. 5b, d). Therefore, *EE*-OBBO shows stronger intermolecular TSI than *EE*-DMODA. Frontier molecular orbitals of *EE*-OBBO were also calculated (Supplementary Fig. 37). The energy gap between HOMO and LUMO of *EE*-OBBO (4.019 eV) was narrower than that of *EE*-DMODA (4.507 eV), which was consistent with the longer emission wavelength of *EE*-OBBO. In addition, the electron distributions of two typical dimers of *EE*-OBBO and *EE*-DMODA selected from their single-crystal structures were also investigated (Fig. 6a, b and Supplementary Fig. 38). There is almost no electron overlap at HOMO but significant overlaps between $n$ and $\pi$ electrons at LUMO in all dimers, suggesting the formation of TSI in the excited state[32]. The more considerable electron overlaps of $n$ and $\pi$ electrons in *EE*-OBBO than those in *EE*-DMODA further indicated its stronger intermolecular TSI. The natural bond orbital (NBO) analysis with second-order perturbation of *EE*-OBBO and *EE*-DMODA was also carried out. The results also revealed that the intermolecular hydrogen bonds and $n\cdots\pi$ interactions are the main reasons for the bright CL (Supplementary Figs. 39 and 40). It is noted that all dimers

do not exist alone. They could form different domains of clusters, as shown in Fig. 6c. Accordingly, small domains with loose packing and large domains with tight packing could further form different clusters with different extents of electron delocalization, corresponding to emitting species with varying energy gaps. That is a widely acknowledged explanation for the excitation-dependent luminescence of clusteroluminogens[32]. In addition, the time-resolved PL decay curves of these clusteroluminogens were measured (Supplementary Figs. 41–47). All their emission lifetimes are below 10 ns, indicating the nature of fluorescence. Therefore, using different starting materials, the intermolecular TSI of clusteroluminogens was successfully manipulated to obtain tunable emission colors and luminescent efficiency. A clear structure-property relationship and isotope effect of clusteroluminogens were built to replenish the TSI mechanism of CL.

## H2O-involved interfacial polymerization

A series of nonaromatic small molecules with tunable emission colors were successfully obtained, but their PLQYs were moderate due to the comparatively weak TSI of small molecules. A potential strategy is to connect small molecules through covalent bonds to promote multiple TSI. Thus, the polymerization method was additionally applied to construct nonaromatic polymers with much higher PLQY. As a concept, PIE has been put forward in recent years, which suggests that the

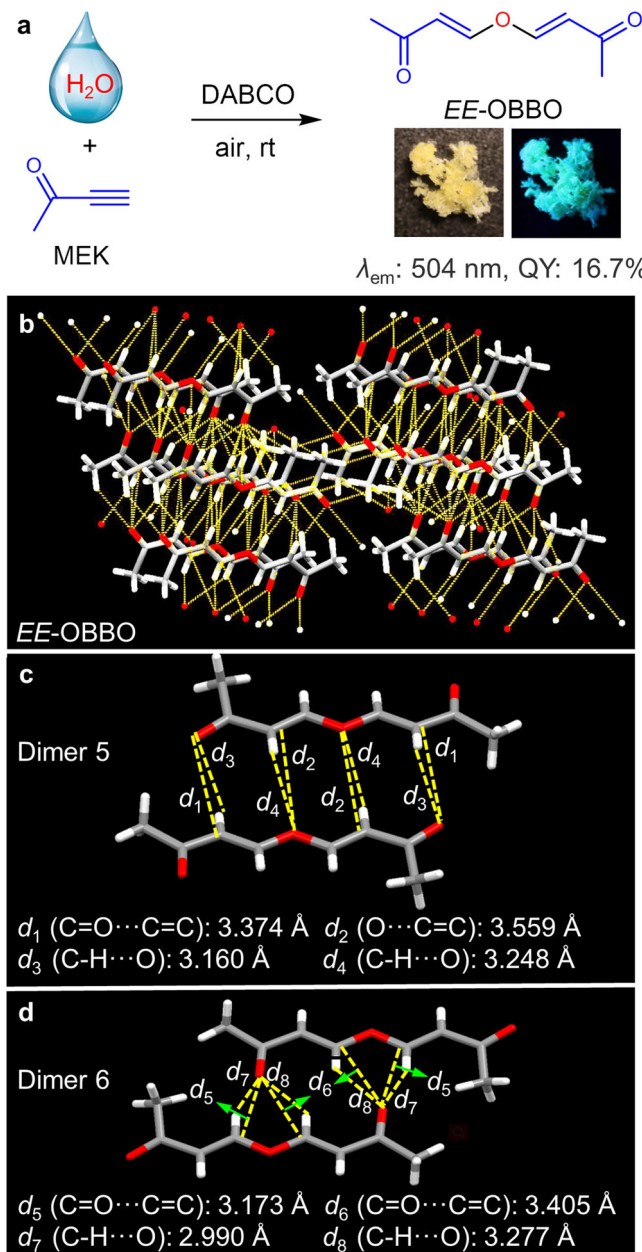

**Fig. 5 | A CLgen from water and ketone-activated alkyne. a** One-step conversion from $H_2O$ and MEK to nonaromatic CLgen of *EE*-OBBO. Photophysical data and photographs of solid powders taken under daylight (left) and UV light (right) are given. **b** Global hydrogen bonds, $n\cdots n$, and $n\cdots\pi$ interactions of *EE*-OBBO. **c**, **d** The $C=O\cdots C=C$, $O\cdots C=C$, and C-H$\cdots$O interactions in different dimers from *EE*-OBBO.

*(Figure 5a labels: $H_2O$ + MEK → DABCO / air, rt → EE-OBBO; $\lambda_{em}$: 504 nm, QY: 16.7%)*

*(Figure 5c, Dimer 5:)* $d_1$ (C=O$\cdots$C=C): 3.374 Å  $d_2$ (O$\cdots$C=C): 3.559 Å  $d_3$ (C-H$\cdots$O): 3.160 Å  $d_4$ (C-H$\cdots$O): 3.248 Å

*(Figure 5d, Dimer 6:)* $d_5$ (C=O$\cdots$C=C): 3.173 Å  $d_6$ (C=O$\cdots$C=C): 3.405 Å  $d_7$ (C-H$\cdots$O): 2.990 Å  $d_8$ (C-H$\cdots$O): 3.277 Å

polymer's emission intensity would increase along with the increased degree of polymerization[15,36]. The reason is that the distances of different groups in polymer chains become shorter when the degree of polymerization increases. Then the clusters with delocalized electrons were formed via enhanced TSI.

Herein, a triyne monomer of TMP was designed and utilized to react with $H_2O$ to generate a nonaromatic polymer, PTMP (Fig. 7a). TMP could be synthesized via simple one-step esterification of trimethylolpropane and propiolic acid[50,51]. Because TMP is hydrophobic and cannot dissolve in water. An interfacial polymerization was carried out, as shown in Fig. 7b. TMP was dissolved in dichloromethane (DCM), and DABCO was dissolved in water. A thin film was formed in the liquid−liquid interface only after a short time. The gel point of this polymerization could be calculated from the Flory Statistics to gain

insight into the polymerization process. Generally, for an $A_2 + B_2 + A_n$ ($n > 2$) monomers system, the Flory Statistics can be simplified into Eq. 1:

$$(p_A)_C = \frac{1}{[r + r\rho(n-2)]^{\frac{1}{2}}} \qquad (1)$$

Where $(p_A)_C$ is the critical reaction extent of group A, $r$ is the mole ratio of group B and A, and $\rho$ is the percentage of group A from monomer $A_n$. Specifically, in our TMP ($A_3$) + $H_2O$ ($B_2$) comonomer system, $\rho = 1$, so Eq. 1 can be further simplified into Eq. 2

$$(p_A)_C = \frac{1}{[2r]^{\frac{1}{2}}} \qquad (2)$$

In $H_2O$/DCM interface, the water concentration is about 55.5 M while the TMP concentration is $10^{-3}$ M, so $r$ in Eq. 2 is large, and $(p_A)_C$ approaches zero. Thus, this polymerization could reach the gel point at the very beginning. The soluble hyperbranched structures could not be obtained. The resultant polymeric film could be free-standing, as shown in Fig. 7c.

The Fourier transform infrared (FT-IR) spectra and $^{13}C$ cross-polarization/magic angle spinning nuclear magnetic resonance (CP/MAS NMR) were used to assist the structural characterization of PTMP. The FT-IR spectra of PTMP and its corresponding monomer TMP and model compound *EE*-DMODA and *EZ*-DMODA are shown in Fig. 8a−d. The absorption bands of TMP associating with the $\equiv C - H$ and $C \equiv C$ stretching vibrations are observed at 3236 and 2112 $cm^{-1}$, respectively. In both the spectra of *EE*-DMODA, *EZ*-DMODA, and PTMP, $\equiv C - H$ and $C \equiv C$ stretching vibration peaks disappeared, revealing the occurrence of interfacial polymerization. Instead, two new stretching vibration peaks of $C = C$ at 1617 $cm^{-1}$ and $C = C - O$ at 1116 $cm^{-1}$ are observed[52]. The $^{13}C$ NMR spectrum of TMP and $^{13}C$ CP/MAS NMR spectrum of PTMP are also given in Fig. 8e, f. The resonance peaks of acetylene carbon of TMP at 74.0 and 75.8 ppm disappeared in the $^{13}C$ CP/MAS NMR spectrum of PTMP. Instead, two new resonance peaks of $C = C$ (f, g) are observed. The resonance peak of ester carbon (a) also shifted from 152.2 to 166.4 ppm. The above results confirm the desired structure of synthesized PTMP. The thermal stability of PTMP was evaluated by thermogravimetric analysis (TGA) and differential scanning calorimeter (DSC). The TGA result indicated that the 5% weight loss temperature ($T_d$) is 268 °C (Supplementary Fig. 48), suggesting they are thermally stable. The DSC result indicated that the glass-transition temperature ($T_g$) is 82 °C (Supplementary Fig. 49), indicating that the generated cross-linked polymer is in a glassy state at room temperature. PTMP is amorphous according to its PXRD pattern (Supplementary Fig. 50). The stress-strain curve for PTMP film (60 min) was obtained via a dynamic thermal mechanical analyzer (DMA) (Supplementary Fig. 51). The Young's modulus is 99.3 MPa, which is comparable to reported cross-linked polymers obtained via the interfacial polymerizations[53].

### Interfacial polymerization-enhanced emission

An interesting phenomenon was observed during this $H_2O$-involved interfacial polymerization. With the extension of polymerization time, the emission wavelength of the polymeric film would red-shift, and the PLQY also increased (Fig. 9a). Herein, a concept named interfacial polymerization-enhanced emission (IPEE) was proposed. The monomer of TMP is weakly emissive with a low PLQY of 2.1%. However, the polymeric film emitted a bright blue light with a higher PLQY of 31.1% when the interfacial polymerization continuously took place for 5 min. After 20 min, a green emissive film could be obtained, and its PLQY also increased to 39.6%. The emission color of resultant films could be red-shifted to yellow, and the highest PLQY could reach 45.7% when

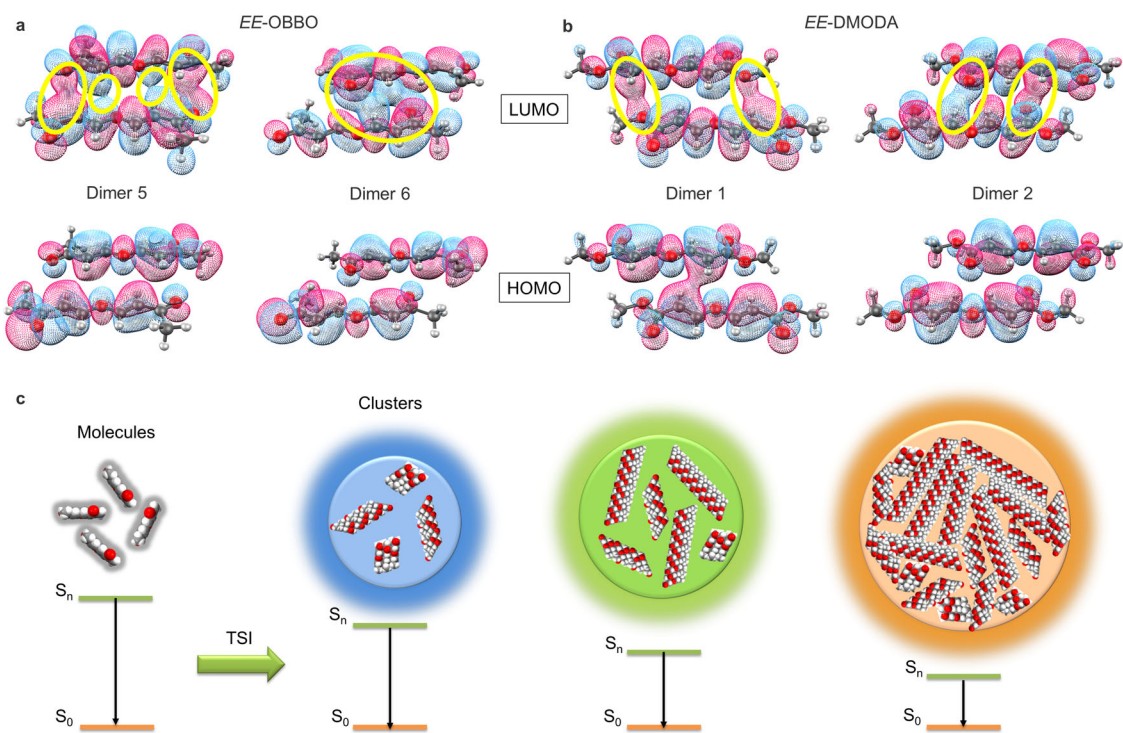

**Fig. 6 | Mechanism study of TSI.** Frontier molecular orbitals of typical dimers of **a** *EE*-OBBO and **b** *EE*-DMODA selected from their single-crystal structures. **c** Schematic illustration of cluster formation accompanied with the corresponding CL properties.

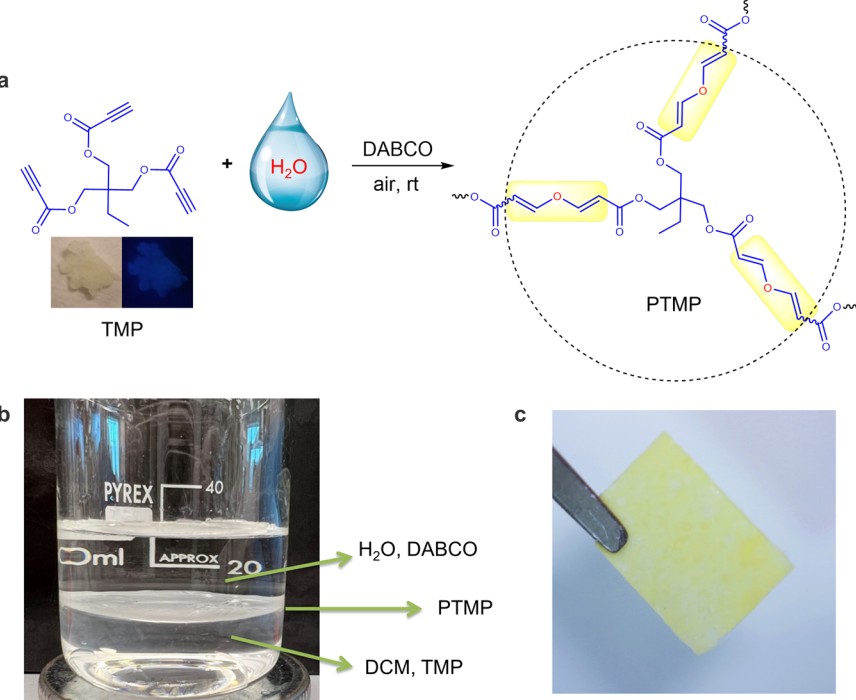

**Fig. 7 | H₂O-involved interfacial polymerization. a** Chemical reaction formula, **b** photograph of interfacial polymerization, and **c** resultant free-standing polymeric film.

the polymerization time was prolonged to 60 min, which is by far the relatively high value among reported clusteroluminogens with long-wavelength emission[12,13]. Further extending the time led to a slightly changed emission color and PLQY. Compared to traditional PIE in solution, this phenomenon of IPEE happens in the liquid–liquid inter-face, which is a strictly confined space. Thus, interfacial polymerization

is more efficient in enhancing TSI than the reported polymerization in solution. All polymeric films showed excitation-dependent lumines-cence (Supplementary Figs. 52–56), and their emission was also con-firmed to be fluorescence, as evidenced by their emission lifetimes (Supplementary Figs. 57–59). Their microscopic morphologies were measured via SEM to determine the difference among resultant

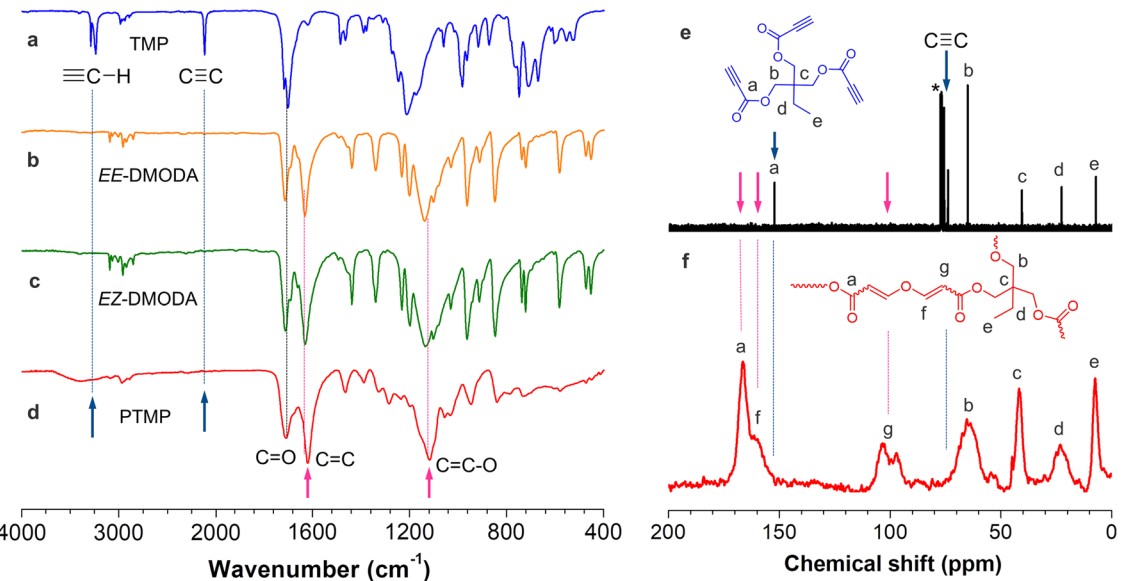

**Fig. 8 | Structural characterization.** FT-IR spectra of **a** monomer TMP, **b** *EE*-DMODA, **c** *EZ*-DMODA, and **d** polymer PTMP. **e** $^{13}$C NMR spectrum of monomer TMP in CDCl$_3$, and **f** $^{13}$C CP/MAS NMR spectrum of polymer PTMP.

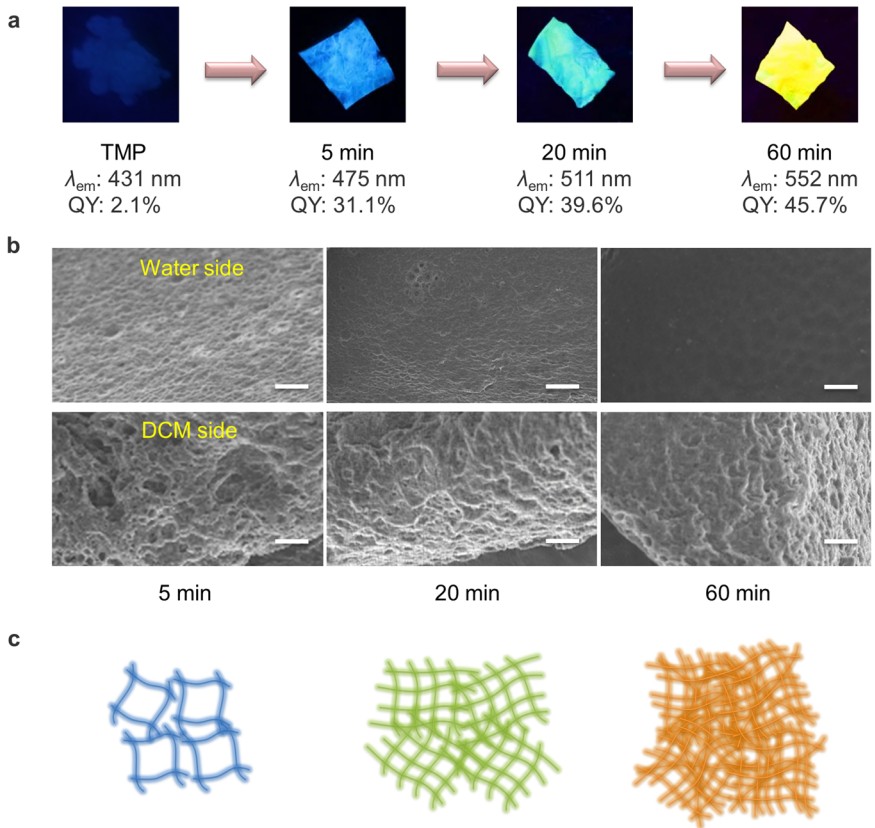

**Fig. 9 | Interfacial polymerization-enhanced emission. a** The photographs of TMP and polymeric films at different polymerization times under UV light. **b** Scanning electron microscope of polymeric films at different polymerization times (scale bar: 20 μm). **c** Schematic diagrams of the polymer structures at different polymerization times.

polymeric films at different reaction times. As shown in Fig. 9b, when the polymerization time was 5 min, two sides of the resultant polymeric film were rough and porous. It indicated that a loose packing of the polymeric film was formed, and crosslinking density was low (Fig. 9c). After 20 min, the surface of the water side of the resultant polymeric film became smoother, and the crosslinking density further increased. When the polymerization time was prolonged to 60 min, a much smoother surface of the water side was obtained. Thus, a remarkably tight packing was obtained, and the crosslinking density was also high. The roughness analysis of atomic force microscope (AFM) images for the polymeric films at different polymerization times was also carried out. Root mean square roughness ($R_q$) and average

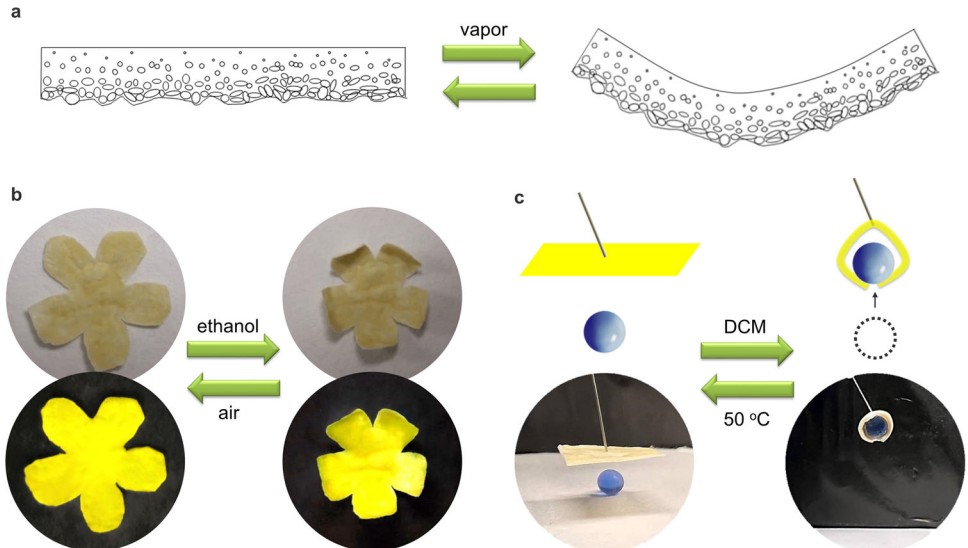

**Fig. 10 | Vapor-responsive soft actuator. a** Schematic illustration of the vapor-triggered bending-recovering behavior of PTMP. **b** Images of a polymer flower with an open-close transformation upon alternate exposure to ethanol vapor or air (up: daylight, down: UV light). **c** Images of a soft mechanical arm (~20 mg) lifting a ball (~80 mg) upon exposure to DCM vapor and releasing it at 50 °C.

roughness ($R_a$) are given by Nanoscope Analysis 1.8 (Supplementary Figs. 60 and 61). The $R_q$ and $R_a$ values decreased both on the water and DCM sides when the polymerization time was prolonged. It is also evidence that tighter structures were formed, and crosslinking density enhanced when the polymerization time was prolonged. Hence, the tight packing and high crosslinking density enhance both intermolecular and intramolecular TSI, promoting long-wavelength emission and much high PLQY of the formed polymeric films.

### Vapor-responsive soft actuator
Mechanically responsive actuators could convert external stimuli (e.g., chemical vapor, light) into reversible mechanical deformations (e.g., bending, rolling, twisting)[54–57]. They have broad applications in many high-tech fields, such as sensors, robots, energy storage, and biomedical engineering[58–60]. These materials usually require elaborate design and multi-step complicated synthesis. Moreover, they do not exhibit luminescence, limiting their applications under dark conditions. Thus, luminescent actuators with simple synthetic routes are highly desired but still challenging.

Many reported clusteroluminogens as solid powder show poor processability. Different from the reported works, herein, we could in situ generate free-standing polymeric films with CL properties via a simple one-step and $H_2O$-involved interfacial polymerization. The resultant polymeric film showed the smooth feature of the water side and the rough character of the DCM side. Accordingly, the resultant polymeric film could be regarded as a Janus film. Two sides of the Janus film exhibited different surface areas and cavities, leading to different vapor absorption capacities. Thus, it could be a mechanically responsive actuator upon chemical vapor (Fig. 10a). Ethanol and DCM vapor were utilized for demonstration, respectively. The Janus film displayed moderate ethanol vapor absorption capacity. As a result, the resultant polymeric film bent slightly after exposure to ethanol vapor (~20 s) and could recover to the initial state when the vapor was removed (~15 s) (Supplementary Fig. 62). This bending-recovering process can be repeated for at least 10 cycles without any deterioration, suggesting its good repeatability. Besides, this Janus film could be cut into different shapes for various potential applications. For example, a film with a flower shape was fabricated. The opened flower could close under ethanol vapor and bloom after removing the vapor (Fig. 10b). This flower emits bright yellow light under UV irradiation, which could be

used in dark conditions. In addition, due to the excellent DCM vapor absorption capacity, this Janus film could bend obviously under DCM vapor (~2 s) and recover to the initial state after heating at 50 °C (~60 s) (Supplementary Fig. 63). Surprisingly, it could serve as a soft robot arm to grab and drop a ball under the control of DCM vapor (Fig. 10c).

## Discussion
In this work, $H_2O$ was successfully utilized as a raw material to construct nonaromatic clusteroluminogens. A reaction of $H_2O$ and alkyne could occur at room temperature under air in the presence of the organic base DABCO. The resultant two products with different conformations, *EE*-DMODA and *EZ*-DMODA, are nonaromatic clusteroluminogens that could emit visible light in the aggregate state. $D_2O$ and $H_2^{18}O$ were further used as starting materials to manipulate the TSI of resultant products to obtain tunable emission colors and luminescent efficiency. Accordingly, the isotope effect for CL was studied systematically. Besides, a CLgen with red-shifted emission and higher luminescence QY was generated using ketone-activated alkyne instead of ester-activated alkyne. Hence, a clear structure-property relationship was built to replenish the TSI mechanism of CL. Due to the easy monomer design, an $H_2O$-involved interfacial polymerization was developed to in situ generate free-standing nonaromatic CL polymeric film with a much higher PLQY (up to 45.7%). An interesting IPEE phenomenon of the synthesized interfacial polymer was observed, where the emission red-shifted and PLQY increased when the polymerization time was prolonged. The resultant polymeric film as a Janus film showed two different surfaces and vapor absorption capacities, exhibiting a vapor-triggered reversible mechanical response that could be applied as a vapor-responsive actuator.

## Methods
### Materials
Methyl propiolate (MP, >98.0%), methyl ethynyl ketone (MEK, >97.0%), 1,4-diazabicyclo[2.2.2]octane (DABCO, >98.0%), and trimethylolpropane (>98.0%) were purchased from Tokyo Chemical Industry Co, Ltd. Heavy water ($D_2O$, 99.9 atom% D) and heavy oxygen water ($H_2^{18}O$, 98 atom % $^{18}O$), and propiolic acid (98%) were purchased from Energy Chemical. THF (inhibitor-free, HPLC grade, ≥99.9%) and DCM (HPLC grade, ≥99.9%) were purchased from Sigma-Aldrich. Water was purified with a Millipore filtration system.

## Instruments

$^1$H and $^{13}$C NMR spectra were measured on a Bruker Avance 400 MHz NMR spectrometer, calibrating chemical shifts using deuterated chloroform (CDCl$_3$) ($^1$H NMR: $\delta$ 7.26 ppm; $^{13}$C NMR: $\delta$ 77.02 ppm) as an internal reference. The IR spectra were measured on a Perkin-Elmer 16 PC FTIR spectrophotometer. High-resolution mass spectra (HRMS) were estimated on a GCT premier CAB048 mass spectrometer operated in a GC-TOF module with chemical ionization (CI). Cross-polarization/magic angle spinning nuclear magnetic resonance (CP/MAS NMR) spectrum was measured on BRUKER AVANCE NEO 400WB. UV-Vis spectra were recorded on a Varian Cary 50 Conc UV-Visible Spectrophotometer with Peltier. PL spectra were recorded on an Edinburgh FS5 Spectrofluorometer. Absolute fluorescence quantum yields were measured on a Hamamatsu Quantum Yield Spectrometer C11347 Quantaurus. Fluorescence lifetime was recorded on an Edinburgh FLS980 Spectrometer. Single-crystal X-ray diffraction (XRD) data were collected on a Rigaku Oxford Diffraction SuperNova with Atlas Diffractometer, and crystal structures were solved with Olex2. Powder X-ray diffraction (PXRD) was collected on an X'per Pro (PANalytical) instrument at 25 °C (scan range: 3–40°). Scanning electron microscope (SEM) was tested by Hitachi Regulus8100. Atomic force microscope (AFM) was tested by Bruker Dimension ICON. Root mean square roughness (R$_q$) and average roughness (R$_a$) are given by Nanoscope Analysis 1.8. Thermogravimetric analysis (TGA) was carried out on a SHIMADZU DTG-60AH analyzer under a nitrogen atmosphere at a heating rate of 10 K/min. Differential scanning calorimeter (DSC) was carried out on a Mettler Toledo DSC3 analyzer under a nitrogen atmosphere at a heating rate of 10 K/min. Elemental analysis (EA) was carried out on a ThermoFisher Scientific FlashSmart CHNS Elemental analyzer. Stress-strain curve was obtained via TA Q800 dynamic thermal mechanical analyzer (DMA). Size measurements were conducted on Dynamic Light Scattering (ZSE, Malvern, UK). All digital photos were recorded on a Canon EOS 60D camera.

## Computational details

All molecules were calculated using the (time-dependent) density functional theory method with B3LYP density functional and 6-31G(d,p) basis set. Grimme's DFT-D3 correction was utilized to better describe London-dispersion effects. Analytical frequency calculations were also performed at the same level of theory to confirm the local minimum point of the optimized structures. To mimic the crystal-packing environment, an ONIOM method was utilized to optimize the structure conformation with the combined quantum mechanics and molecular mechanics (QM/MM) approach. The calculated molecular models (dimers) were extracted from their corresponding single-crystal structures with the CCDC numbers 2177902 (*EE*-DMODA), 2177911 (*EZ*-DMODA), and 2177919 (*EE*-OBBO). The central two molecules (dimer) were treated as the QM part and optimized at the B3LYP-D3/6-31 G(d,p) level, and the surrounding molecules were frozen as the MM part with the universal force field (UFF). Natural bond orbital (NBO) analysis was performed based on the optimized dimer structures, which were displayed using Multiwfn software (version 3.8)[61]. All calculations were performed using Gaussian 16 (Revision A.03) program. The HOMO and LUMO electron distributions were visualized by IQmol software (version 3.0).

## Procedures for the reaction of H$_2$O and activated alkyne MP

MP (445 μL, 5 mmol), DABCO (56 mg, 0.5 mmol), 5 mL H$_2$O, and 1 mL THF were placed into a 10-mL tube equipped with a magnetic stir bar. The mixture was stirred at 25 °C for 24 h under air. After solvent evaporation, the crude product was purified by a silica gel column using hexane/ethyl acetate (4:1, *v/v*) as eluent (R$_f$ for *EE*-DMODA: 0.4; R$_f$ for *EZ*-DMODA: 0.3). The products *EE*-DMODA (242.2 mg) and *EZ*-DMODA (10.3 mg) were obtained in 54% yield.

*EE*-DMODA: $^1$H NMR (400 MHz, CDCl$_3$) $\delta$ (TMS, ppm): 7.58 (d, $J = 12.0$ Hz, 2H), 5.66 (d, $J = 12.0$ Hz, 2H), 3.74 (s, 6H). $^{13}$C NMR (100 MHz, CDCl$_3$), $\delta$ (ppm): 166.48, 157.39, 103.98, 51.66. ESI-MS: m/z calculated for [M + H]$^+$ C$_8$H$_{11}$O$_5$: 187.0606, found 187.0611.

*EZ*-DMODA: $^1$H NMR (400 MHz, CDCl$_3$) $\delta$ (TMS, ppm): 7.57 (d, $J = 12.4$ Hz, 1H), 6.70 (d, $J = 7.2$ Hz, 1H), 5.71 (d, $J = 12.4$ Hz, 1H), 5.21 (d, $J = 7.2$ Hz, 1H), 3.78 (s, 6H). $^{13}$C NMR (100 MHz, CDCl$_3$), $\delta$ (ppm): 166.71, 164.13, 158.44, 152.34, 103.37, 101.85, 51.64, 51.48. ESI-MS: m/z calculated for [M + H]$^+$ C$_8$H$_{11}$O$_5$: 187.0606, found 187.0605.

## Procedures for the reaction of H$_2$$^{18}$O and activated alkyne MP

MP (445 μL, 5 mmol), DABCO (56 mg, 0.5 mmol), 5 mL H$_2$$^{18}$O, and 1 mL THF were placed into a 10 mL tube equipped with a magnetic stir bar. The mixture was stirred at 25 °C for 24 h under air. After solvent evaporation, the crude product was purified by a silica gel column using hexane/ethyl acetate (4:1, *v/v*) as eluent (R$_f$ for *EE*-DMODA-$^{18}$O: 0.4; R$_f$ for *EZ*-DMODA-$^{18}$O: 0.3). The products *EE*-DMODA-$^{18}$O (172.3 mg) and *EZ*-DMODA-$^{18}$O (21.4 mg) were obtained in 41% yield.

*EE*-DMODA-$^{18}$O: $^1$H NMR (400 MHz, CDCl$_3$) $\delta$ (TMS, ppm): 7.58 (d, $J = 12.0$ Hz, 2H), 5.66 (d, $J = 12.0$ Hz, 2H), 3.74 (s, 6H). $^{13}$C NMR (100 MHz, CDCl$_3$), $\delta$ (ppm): 166.48, 157.38, 103.94, 51.66. ESI-MS: m/z calculated for [M + H]$^+$ C$_8$H$_{11}$O$_4$$^{18}$O: 189.0732, found 189.0770.

*EZ*-DMODA-$^{18}$O: $^1$H NMR (400 MHz, CDCl$_3$) $\delta$ (TMS, ppm): 7.57 (d, $J = 12.0$ Hz, 1H), 6.71 (d, $J = 6.8$ Hz, 1H), 5.72 (d, $J = 12.0$ Hz, 1H), 5.21 (d, $J = 6.8$ Hz, 1H), 3.74 (s, 6H). $^{13}$C NMR (100 MHz, CDCl$_3$), $\delta$ (ppm): 166.65, 164.29, 158.42, 152.35, 103.36, 101.81, 51.64, 51.49. ESI-MS: m/z calculated for [M + H]$^+$ C$_8$H$_{11}$O$_4$$^{18}$O: 189.0732, found 189.0751.

## Procedures for the reaction of D$_2$O and activated alkyne MP

MP (445 μL, 5 mmol), DABCO (56 mg, 0.5 mmol), 5 mL D$_2$O, and 1 mL THF were placed into a 10 mL tube equipped with a magnetic stir bar. The mixture was stirred at 25 °C for 24 h under air. After solvent evaporation, the crude product was purified by a silica gel column using hexane/ethyl acetate (4:1, *v/v*) as eluent (R$_f$ for *EE*-DMODA-D$_4$: 0.4; R$_f$ for *EZ*-DMODA-D$_4$: 0.3). The products *EE*-DMODA-D$_4$ (199.8 mg) and *EZ*-DMODA-D$_4$ (19.9 mg) were obtained in 46% yield.

*EE*-DMODA-D$_4$: $^1$H NMR (400 MHz, CDCl$_3$) $\delta$ (TMS, ppm): 7.57 (s, 0.14H), 5.64 (s, 0.23H), 3.72 (s, 6H). $^{13}$C NMR (100 MHz, CDCl$_3$), $\delta$ (ppm): 166.45, 157.29, 103.80, 51.62. ESI-MS: m/z calculated for [M + H]$^+$ C$_8$H$_7$D$_4$O$_5$: 191.0858, found 191.0912.

*EZ*-DMODA-D$_4$: $^1$H NMR (400 MHz, CDCl$_3$) $\delta$ (TMS, ppm): 7.57 (s, 0.12H), 6.71 (s, 0.06H), 5.70 (s, 0.12H), 5.20 (s, 0.07H), 3.74 (s, 6H). $^{13}$C NMR (100 MHz, CDCl$_3$), $\delta$ (ppm): 166.69, 164.14, 158.37, 152.43, 103.16, 101.69, 51.63, 51.45. ESI-MS: m/z calculated for [M + H]$^+$ C$_8$H$_7$D$_4$O$_5$: 191.0858, found 191.0905.

## Procedures for the reaction of H$_2$O and activated alkyne MEK

MEK (391 μL, 5 mmol), DABCO (56 mg, 0.5 mmol), and 5 mL H$_2$O were placed into a 10 mL tube equipped with a magnetic stir bar. The mixture was stirred at 25 °C for 24 h under air. After solvent evaporation, the crude product was purified by a silica gel column using hexane/ethyl acetate (2:1, *v/v*) as eluent (R$_f$ for *EE*-OBBO: 0.2). The products *EE*-OBBO (123.2 mg) was obtained in 32% yield.

*EE*-OBBO: $^1$H NMR (400 MHz, CDCl$_3$) $\delta$ (TMS, ppm): 7.52 (d, $J = 12.0$ Hz, 2H), 6.00 (d, $J = 12.0$ Hz, 2H), 2.24 (s, 6H). $^{13}$C NMR (100 MHz, CDCl$_3$), $\delta$ (ppm): 196.44, 156.83, 113.08, 28.80. ESI-MS: m/z calculated for [M-H]$^+$ C$_8$H$_9$O$_3$: 153.0552, found 153.0551.

## Data availability

The authors declare that all the data supporting the findings of this manuscript are available within the manuscript and Supplementary Information files and available from the corresponding authors upon request. Crystallographic data for the structures reported in this article have been deposited at the Cambridge Crystallographic Data Centre

**Article** https://doi.org/10.1038/s41467-023-38769-y

under deposition numbers CCDC 2177902 (*EE*-DMODA), 2177911 (*EZ*-DMODA), 2177917 (*EE*-DMODA-¹⁸O), and 2177919 (*EE*-OBBO). Copies of the data can be obtained free of charge via https://www.ccdc.cam.ac.uk/structures/.

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

## Acknowledgements

This work was supported by the National Natural Science Foundation of China Grant (21788102), the Research Grants Council of Hong Kong (16307020, C6014-20W, and N_HKUST609/19), the Innovation and Technology Commission (ITC-CNERC14SC01), and the Natural Science Foundation of Guangdong Province (2019B121205002 and 2019B030301003). The authors would like to thank the Shiyanjia lab (www.shiyanjia.com) for the CP/MAS NMR test.

## Author contributions

B.S. and B.Z.T. conceived and designed the experiments. B.S. performed the synthesis. B.S. and J.Y.Z. did the photophysical measurements and analyzed the data. J.Y.Z. conducted theoretical calculations. J.D.Z. performed the single-crystal measurements. B.S., J.Y.Z., A.Q., J.W.Y.L., and B.Z.T. took part in the discussion and gave important suggestions. B.S. and J.Y.Z. co-wrote the paper.

## Competing interests

The authors declare no competing interests.
