## [Peer Review File · Nature Communications]

Facile conversion of water to functional molecules and cross-linked polymeric films with efficient clusteroluminescenceReviewers' Comments:

Reviewer #1:

Remarks to the Author:

The submitted communication by Tang and coworkers describes an interesting oligomerization/cross-linking reaction involving water as a building block. Running the reaction at a water/DCM solvent interface yields a cross-linked network material which has different morphologies on either side of the film, leading to a reversible film "curling" response in the presence of certain solvents. Likewise, emission enhancement is seen as the material becomes more cross-linked/rigid, in line with a type of aggregation-enhanced emission. Overall, this work reaches the level of novelty required for publication in Nature Comm., however, I have several major corrections that need to be addressed prior to publication.

Major concerns:

- The authors seem fond of using the term "polymerization-induced emission" or PIE to describe the luminescence in this work. However, the alkyne monomers themselves are weakly luminescent, so there is no "induction" of emission upon reaction with water, but an enhancement of emission.

On a related point, the authors use a change in emission profiles relative to excitation wavelength as a hallmark of "clusterluminescence" or CL. Isn't this just aggregation-enhanced emission? There seems to be unnecessary "rebranding" of emission properties to sell their work here. On a related point, the authors should track the evolution of "CL" by performing dynamic light scattering to determine if an aggregate size increase is commensurate with red-shifted emission.

- I found much of the initial work on the monomers and the explanation of subtle (reproducible?) differences in the emission a bit cumbersome to read through. The authors should summarize the quantum yield, emission maxima, and lifetime data for each system in a Table. Given that the emission of the monomers is very weak, are the authors sure that any differences noted in emission quantum yield are outside of the standard error/reproducibility of their measurements?

- I also found the identification of somewhat close O---O atomic separations as evidence for intermolecular interactions unconvincing. For example, one would need a type of O(lone pair) to O-E (E = element) σ^* or π^* orbital interaction to validate such a postulate. Do the authors see any such interactions by performing second order perturbation analysis with NBO computations? In fact, it seems to me that hydrogen bonding is the main mechanism by which small molecule aggregation and restriction of intramolecular rotation (RIR)/emission enhancement is happening.

- The authors must, and I restate must, provide full details related to the preparation of any required starting materials used in synthesis, sources/degree of isotope substitution for all labelled reagents, and even more critically, FULL details regarding the acquisition of luminescence data (including equipment used) and methods (basis sets, functionals, etc...) for the computational investigations. Right now, the only computational details in the ESI and xyz coordinates, yet it is clear that a wide range of computational investigations were done. Moreover, the authors need to provide information of the fitting of lifetime data.

Minor comments:

- The formation of the deuterated (D4) "dimers" outlined in Fig 4c is depicted in a misleading way. The authors use a very large excess of D2O in this reaction, yet according to the expected balanced equation one would expect D2-isotopomers in this process. So, this means that some added C-H/D2O exchange is occurring to yield extra deuteration. What happens when this reaction is conducted in a 1:2 D2O/alkyne ratio? Are the expected D2 isotopomers formed?

- I found the extended discussion on how green water is (as a reagent) irrelevant to the current study, since DABCO is needed to promote the reaction. Likewise, I assume that the alkynes are not produced in a very "green" way.

- Abstract: "different water and alkyne" does not make sense to me. How can have different water? Do the authors mean "by reacting water with different alkynes".

- Title: The term "interfacial-polymers" is misleading/confusing to me. Do the authors mean polymers that are made using interfacial polymerization? If so, then perhaps the title should be updated as "...to Functional Cross-Linked Materials via Interfacial Polymerization with...". When I first read "interfacial polymers" I was not sure if the authors meant a linear polymer made by interfacial polymerization, such as nylon, or if they meant a block copolymer was being investigated, which can phase separate to yield heterogenous interfaces.

- Page 2, line 4: "Most species from water..." is an incomplete statement. Do the authors mean, most species prepared using water as a reactant..."

- Page 2, line 8: replace "fluorine" by "fluorene"

- Page 2, paragraph 3 (and elsewhere): I object to the term "artificial" to describe any chemical reaction that is not conducted biochemically. The root of this word, "artifice" means to deceive or trick. There is no trickery here, just a reaction that involves base-assisted addition of water to an alkyne.

- Page 5, lines 2-3: "...an intermolecular O---O distance of 2.883 Å in..."; see comment above. Moreover, all reported bond lengths and angles that are determined by X-ray crystallography should be given with their estimated standard deviations (esds).

- Page 5, paragraph 1: "narrower energy gap"; "lower unoccupied molecular orbital (LUMO) [energies]". Please give actual numbers to support your discussions here.

- Much of the discussion related to how isotope substitution would influence the degree/strength of intermolecular interactions, such as hydrogen bonding, is not described well here. The authors are trying to rely on minor changes in emission data and XRD profiles to make fairly general statements. For example, I do not see how partial deuteration in a molecule would lead to difficulties in crystallization? Has this phenomenon been observed by others? There is a dearth of references to literature work in this section, to the point where it reads like "trust us" instead of a convincing argument for the role of isotopes in modifying the rigidity of a molecular framework in the solid state and thus its emission.

- Page 8, line 1: "significant [overlap] between pi and sigma electrons at LUMO" does not make any sense to be. A LUMO is an unoccupied orbital... Later the authors say "more considerable [overlap]", how was this judged? By eye? Or by measurable parameters?

- Page 9: I am wondering why the authors did not use a diyne building block in their work? This should yield a soluble/processable linear polymer.

- Page 10, line 4: "the water concentration would be infinite". No. Pure water has a concentration of approximately 1000/18.01 M or 55.5 M under STP.

- Page 13: How heavy was the "drop ball"?

- ESI: Table 1 requires R1, wR2 and GOF values for the X-ray data presented. Otherwise, it is difficult for others to determine the quality of the X-ray refinement procedures.

- ESI: How were the samples for the solid PL measurements prepared? Drop-cast from solution? etc..?
- ESI: Rf values should be given for all species purified by column chromatography.
- Fig S8 and S32: I am wondering why the authors did not use TD-DFT to probe the excitations leading to emission? Use of TD-FT is standard in the community these days.
- ESI: It is odd that "56 mg" of alkyne was used in each reaction? Was this on purpose or perhaps a copy/paste error?

Reviewer #2:

Remarks to the Author:

This manuscript describes the formation of clusteroluminescent compounds (CLgens) and polymers through a reaction of water molecules and alkynes in the presence of DABCO as an organic catalyst. Careful screening of the reaction using different types of water molecules such as heavy oxygen water (H_2^{18}O) and heavy water (D_2O) can suggest water involved reaction for the CLgen synthesis and the resulting molecular packing structures in their crystalline/solid-state control the through-space interactions (TSI). The small molecular CLgens' synthesis discussion is clear and well written. In this study, the CLgens' synthesis was applied at a water/organic solvent interface, and in the case of TMP as an alkyne reactant, the resulting product was 3D networking polymers. The new phenomenon of interfacial polymerization-induced emission (IPIE) sounds attractive. However, this reviewer claims the investigation of the polymer products. For the structure analysis, IR and ^{13}C CP/MAS were utilized. How about further analysis by means of XPS, elemental analysis, XRD, TGA, and DSC? There are discussions about crosslinking density of the polymeric films which depends on the polymerization time, however, no quantitative investigations such as porosity, surface area, density, etc, were carried out. Even AFM analysis, how are AFM images of 5 min and 20 min polymerization conditions? It may be possible to discuss more strictly the change of roughness evaluated from AFM. Is it really increasing the crosslinking (chemically) density or just aggregating the polymer and forming thicker films? In order to know this point, this reviewer recommends evaluating the polymer films by mechanical testing, such as a tensile test or bending test. Are they comparable to the 3D networking conventional polymers or elastomers? In the discussion about the actuator properties, I wonder about the responding speed and mechanical forces. Without such quantitative values, we are hard to imagine the performance and ability of the actuators.

Overall, although this type of reaction is brand-new and attractive using water as a reactant to design nonaromatic fluorescent and vapor-responsive materials, I judge this study needs further investigation in particular the polymers produced at the water/organic solvent interface. Other minor suggestions are listed below.

1) In the time-resolved PL decay analysis, Figs. 34-38 and Figs. 44-46, a curve of the instrument response function should be added in each figure. The amplitude-weighted lifetime components and fitting factor should be indicated as an accuracy explanation.

2) "First" at the beginning of the Results is a typo.

3) It would be great if the authors study in-situ IPIE detection. How the emission change under growing the polymer film at the interface occur is interesting to know.

Point-by-Point Response to Reviewers' Comments

MS ID.: NCOMMS-23-01434
MS Title: Facile conversion of water to functional molecules and cross-linked polymeric films with efficient clusteroluminescence

Response to the comments and suggestions of Reviewer 1

The reviewer commented that “The submitted communication by Tang and coworkers describes an interesting oligomerization/cross-linking reaction involving water as a building block. Running the reaction at a water/DCM solvent interface yields a cross-linked network material which has different morphologies on either side of the film, leading to a reversible film "curling" response in the presence of certain solvents. Likewise, emission enhancement is seen as the material becomes more cross-linked/rigid, in line with a type of aggregation-enhanced emission. Overall, **this work reaches the level of novelty required for publication in Nature Comm.**, however, I have several major corrections that need to be addressed prior to publication.”

We sincerely thank the reviewer for the strong support of the main findings in this study and constructive suggestions that helped us to further the quality.

1. *The reviewer commented that “The authors seem fond of using the term "polymerization-induced emission" or PIE to describe the luminescence in this work. However, the alkyne monomers themselves are weakly luminescent, so there is no "induction" of emission upon reaction with water, but an enhancement of emission..”*

Response: Thanks the reviewer for this comment. We have changed “interfacial polymerization-induced emission (IPIE)” to “interfacial polymerization-enhanced emission (IPEE)” as the reviewer suggested in the revised manuscript.

2. *The reviewer commented that “On a related point, the authors use a change in emission profiles relative to excitation wavelength as a hallmark of "clusterluminescence" or CL. Isn't this just aggregation-enhanced emission? There seems to be unnecessary "rebranding" of emission properties to sell their work here. On a related point, the authors should track the evolution of "CL" by performing dynamic light scattering to determine if an aggregate size increase is commensurate with red-shifted emission.”*

Response: Thanks the reviewer for this comment. It is acknowledged that clusteroluminescence (CL) is one typical type of aggregation-enhanced emission (AIE) effect. Traditional AIE phenomenon mainly focuses on the largely conjugated molecules and polymers with intrinsic luminescent properties based on through-bond conjugation. However, CL mainly highlighted the luminescence from nonaromatic or nonconjugated molecules and polymers, which originate from through-space interactions in the aggregate state and exhibited longer emission wavelength than their intrinsic level of conjugation (see *JACS Au* **2021**, *1*, 1805-1814; *Chem. Soc. Rev.* **2021**, *50*, 12616-12655; *J. Am. Chem. Soc.* **2021**, *143*, 25, 9565- 9574; *Nat. Commun.* **2022**, *13*, 3492). Thus, CL was a more appropriate term to describe the special luminescent system studied in this work.

We used the dynamic light scattering (DLS) technique to track the evolution of clusters as the reviewer suggested. The large size of clusters could be detected by DLS when the concentration of *EE*-OBBO exceeded critical cluster concentration (CCC). However, no signal was detected when the concentration was lower than the CCC point. Moreover, it was observed that the size of the clusters increased when the concentration increased from 10 to 25 mM, resulting in red-shifted emission and enhanced intensity (Supplementary Figs. 33-36 as shown below). Accordingly, we have added the above statement in the “Manipulation of TSP” section on page 8 of the revised manuscript.

Supplementary Figure 33. Dynamic light scattering diagram of *EE*-OBBO in THF, [*EE*-OBBO] = 10 mM.

Supplementary Figure 34. Dynamic light scattering diagram of *EE*-OBBO in THF, [*EE*-OBBO] = 15 mM.

Supplementary Figure 35. Dynamic light scattering diagram of *EE*-OBBO in THF, [*EE*-OBBO] = 20 mM.

Supplementary Figure 36. Dynamic light scattering diagram of *EE*-OBBO in THF, [*EE*-OBBO] = 25 mM.

3. *The reviewer commented that “I found much of the initial work on the monomers and the explanation of subtle (reproducible?) differences in the emission a bit cumbersome to read through. The authors should summarize the quantum yield, emission maxima, and lifetime data for each system in a Table. Given that the emission of the monomers is very weak, are the authors sure that any differences noted in emission quantum yield are outside of the standard error/reproducibility of their measurements?”*

Response: Thanks the reviewer for this comment. We have summarized the quantum yield, emission maxima, and lifetime data in Supplementary Table 1. We used an integrating sphere to measure the absolute photoluminescence quantum yield (PLQY) on Hamamatsu Quantum Yield Spectrometer C11347 Quantaaurus. The PLQYs of all molecules are much higher than the instrument background noise. Moreover, the PLQY recorded in Supplementary Table 1 were the average value based on three times of measurements.

Supplementary Table 1. Photophysical properties of CLgens.

	λ_{em} (nm) ^a	Φ (%) ^b	τ_{avg} (nm) ^c
EE -DMODA	445	12.9	3.7
EZ -DMODA	478	4.5	2.1
EE -DMODA- ¹⁸ O	485	20.6	3.5
EZ -DMODA- ¹⁸ O	493	10.4	1.8
EE -DMODA-D ₄	437	6.5	3.0
EZ -DMODA-D ₄	452	3.1	1.9
EE -OBBO	504	16.7	3.5

^a λ_{em} = emission maximum in the solid state (λ_{ex} = 380 nm). ^b Absolute fluorescence quantum yield, measured by an integrating sphere. ^c The amplitude-weighted lifetime in the solid state (λ_{ex} = 365 nm). They were calculated from $\tau_{avg} = \tau_1 * Rel_1\% + \tau_2 * Rel_2\% + \tau_3 * Rel_3\%$.

4. *The reviewer commented that “I also found the identification of somewhat close O---O atomic separations as evidence for intermolecular interactions unconvincing. For example, one would need a type of O(lone pair) to O-E (E = element) sigma* or pi* orbital interaction to validate such a postulate. Do the authors see any such interactions by performing second order perturbation analysis with NBO computations? In fact, it seems to me that hydrogen bonding is*

the main mechanism by which small molecule aggregation and restriction of intramolecular rotation (RIR)/emission enhancement is happening.”

Response: Thanks the reviewer for this constructive comment. According to the suggestion, we have conducted the NBO analysis of *EE*-OBBO and *EE*-DMODA, respectively (Supplementary Figs. 39, 40). The results showed that no intermolecular O...O interaction (e.g., O (lone pair) to O-E (E = element)) was observed. Instead, multiple hydrogen bonding interactions and n-pi interactions (C=O ..C=C and O ..C=C) were observed. These results verified that the intermolecular hydrogen bondings and n-pi interactions (C=O ..C=C and O ..C=C) played an essential role in the RIM mechanism and clusteroluminescence as the reviewer proposed. Therefore, we have deleted these intermolecular O...O distances in Figs. 3, 4, and 5 and revised the related statements in the revised manuscript.

Accordingly, we have highlighted the role of hydrogen bonds and n-pi interactions in the revised manuscript that *“The natural bond orbital (NBO) analysis with second-order perturbation of EE-OBBO and EE-DMODA was also carried out. The results also revealed that the intermolecular hydrogen bonds and n ..π interactions are the main reasons for the bright CL (Supplementary Figs. 39, 40).”*

Supplementary Figure 39. Natural bond orbital (NBO) analysis with second-order perturbation of typical dimers of *EE*-OBBO. LP = lone pair, BD* = anti-bonding, $E(2)$ = the stabilization energy with second-order perturbation (unit = kcal/mol).

Supplementary Figure 40. Natural bond orbital (NBO) analysis with second-order perturbation of typical dimers of *EE*-DMODA. LP = lone pair, BD* = anti-bonding, $E(2)$ = the stabilization energy with second-order perturbation (unit = kcal/mol).

5. *The reviewer commented that “The authors must, and I restate must, provide full details related to the preparation of any required starting materials used in synthesis, sources/degree of isotope substitution for all labelled reagents, and even more critically, FULL details regarding the acquisition of luminescence data (including equipment used) and methods (basis sets, functionals, etc...) for the computational investigations. Right now, the only computational details in the ESI and xyz coordinates, yet it is clear that a wide range of computational investigations were done. Moreover, the authors need to provide information of the fitting of lifetime data.”*

Response: Thanks the reviewer for this suggestion. We have added more detailed information in the “Methods” section on pages 15 and 16 in the revised manuscript. We also provide information on the fitting of lifetime data in Supplementary Figs. 41-47 and 57-59.

6. *The reviewer commented that “The formation of the deuterated (D4) “dimers” outlined in Fig 4c is depicted in a misleading way. The authors use a very large excess of D₂O in this reaction, yet according to the expected balanced equation one would expect D₂-isotopomers in this process. So, this means that some added C-H/D₂O exchange is occurring to yield extra deuteration. What happens when this reaction is conducted in a 1:2 D₂O/alkyne ratio? Are the expected D₂ isotopomers formed?”*

Response: Thanks the reviewer for this comment. We have revised Fig. 4d and added more statements to make it easier to understand. Because a vast excess of D₂O was used in this reaction and the ethynyl proton of MP is highly active, it could exchange with D in D₂O. Next, the deuterated MP reacted with D₂O to

generate *EE*-DMODA-D₄ and *EZ*-DMODA-D₄, respectively. According to their ¹H NMR spectra (Supplementary Fig. 22), D₄ isotopomers are the main products. This reaction strictly requires a significant excess of water to obtain the products. According to the suggestion from the reviewer, we also conducted the reaction in a 1:2 D₂O/alkyne ratio. However, no expected D₂ isotopomers could form.

7. *The reviewer commented that "I found the extended discussion on how green water is (as a reagent) irrelevant to the current study, since DABCO is needed to promote the reaction. Likewise, I assume that the alkynes are not produced in a very "green" way."*

Response: Thanks the reviewer for this consideration. According to the suggestion, we just described these new H₂O-involved reaction and interfacial polymerization and did not mention how green these reactions were in the revised manuscript.

8. *The reviewer commented that "Abstract: "different water and alkyne" does not make sense to me. How can have different water? Do the authors mean "by reacting water with different alkynes"."*

Response: Thanks the reviewer for this comment. "Different water" means H₂O, H₂¹⁸O and D₂O. We have changed the statements to "*Their emission colors and luminescent efficiency could be adjusted by manipulating through-space interaction using different starting materials.*" in the revised abstract to avoid misunderstanding.

9. *The reviewer commented that "Title: The term "interfacial-polymers" is misleading/confusing to me. Do the authors mean polymers that are made using interfacial polymerization? If so, then perhaps the title should be updated as "...to Functional Cross-Linked Materials via Interfacial Polymerization with...". When I first read "interfacial polymers" I was not sure if the authors meant a linear polymer made by interfacial polymerization, such as nylon, or if they meant a block copolymer was being investigated, which can phase separate to yield heterogenous interfaces."*

Response: Thanks the reviewer for this comment. In this work, we fabricated functional cross-linked polymers via interfacial polymerization as shown in Figs. 7 and 9. Therefore, we have changed the title to "*Facile conversion of water to functional molecules and cross-linked polymeric films with efficient clusteroluminescence*" in the revised manuscript to avoid misunderstanding.

10. *The reviewer commented that “Page 2, line 4: “Most species from water...” is an incomplete statement. Do the authors mean, most species prepared using water as a reactant...”*

Response: Thanks the reviewer for this comment. According to the suggestion, we have changed the statement to *“Most species prepared using water as a reactant are nonaromatic and nonconjugated, serving as commodities for everyday use.”* on page 2 of the revised manuscript.

11. *The reviewer pointed out that “Page 2, line 8: replace “fluorine” by “fluorene”*

Response: Thanks the reviewer for pointing it out. We have revised it on page 2 in our revised manuscript.

12. *The reviewer commented that “Page 2, paragraph 3 (and elsewhere): I object to the term “artificial” to describe any chemical reaction that is not conducted biochemically. The root of this word, “artifice” means to deceive or trick. There is no trickery here, just a reaction that involves base-assisted addition of water to an alkyne.”*

Response: Thanks the reviewer for this suggestion. We have deleted “artificial” in the revised manuscript.

13. *The reviewer commented that “Page 5, lines 2-3: “...an intermolecular O...O distance of 2.883 Å in...”; see comment above. Moreover, all reported bond lengths and angles that are determined by X-ray crystallography should be given with their estimated standard deviations (esds).”*

Response: Thanks the reviewer for this comment. Here, we highlighted the intramolecular O...O interaction of *EZ*-DMODA with a distance of 2.883 Å, which was determined from the X-ray crystallography as shown in Fig. 2a. The electrostatic potential surface of *EZ*-DMODA also indicated such interaction with a negative potential region between these two oxygens atoms (Fig. 2c). However, no such an interaction was observed for its isomer of *EE*-DMODA. This interaction should be responsible for the red-shifted emission wavelength of *EZ*-DMODA compared with *EE*-DMODA. According to the suggestion from the reviewer, we have added all bond lengths and angles determined by X-ray crystallography with their estimated standard deviations in the “Additional Data” section in our revised Supplementary Information.

14. *The reviewer commented that “Page 5, paragraph 1: “narrower energy gap”; “lower unoccupied molecular orbital (LUMO) [energies]”. Please give actual numbers to support your discussions here.”*

Response: Thanks the reviewer for this comment. We have given actual numbers on page 5 in the revised manuscript.

“The result indicates that EZ-DMODA processes a narrower energy gap (4.427 eV) between the highest occupied molecular orbital (HOMO) and the lowest unoccupied molecular orbital (LUMO) than that of EE-DMODA (4.507 eV), which was consistent with the experimental results.”

15. *The reviewer commented that “Much of the discussion related to how isotope substitution would influence the degree/strength of intermolecular interactions, such as hydrogen bonding, is not described well here. The authors are trying to rely on minor changes in emission data and XRD profiles to make fairly general statements. For example, I do not see how partial deuteration in a molecule would lead to difficulties in crystallization? Has this phenomenon been observed by others? There is a dearth of references to literature work in this section, to the point where it reads like “trust us” instead of a convincing argument for the role of isotopes in modifying the rigidity of a molecular framework in the solid state and thus its emission.”*

Response: Thanks the reviewer for this comment. We have cited many references about the isotope effect of different kinds of luminogens (ref. 44-49). Our molecules are nonaromatic, which are totally different from the reported luminogens. Thus, the isotope effect of CLgens was investigated for the first time. For the deuterated compounds, the deuterated efficiency was deduced to be 91 atom% according to the integration of different peaks in their ¹H NMR spectra (Supplementary Fig. 22). PXRD patterns (some parameters, such as intensity, FWHM, etc.) are always used to determine the quality of the crystals (see *J. Am. Chem. Soc.* **2015**, *137*, 3241–3247; *J. Am. Chem. Soc.* **2017**, *139*, 18322–18327; *World J. Nano Sci. Eng.* **2012**, *2*, 154-160; *J. Chem. Sci.* **2006**, *118*, 127–133; *Cellulose* **2017**, *24*, 1971–1984; *Clays Clay Miner.* **1997**, *45*, 461-475.). Under the same preparation method, the deuterated CLgens show poorer crystallinity according to the PXRD patterns (Supplementary Figs. 24, 25). Thus, “the partial deuteration leads to poorer crystallinity” could be concluded according to our experiments. The reason is that the crystallization driving force of our molecules is regular hydrogen bonds. Due to the incomplete deuterated efficiency, many different compounds with irregular structures were obtained (Supplementary Fig. 23), which made their crystallization difficult. Because our molecules are totally different from the reported luminogens, this phenomenon might be observed for the first time to our best knowledge.

16. *The reviewer commented that “Page 8, line 1: "significant [overlap] between pi and sigma electrons at LUMO" does not make any sense to be. A LUMO is an unoccupied orbital... Later the authors say "more considerable [overlap]", how was this judged? By eye? Or by measurable parameters?”*

Response: Thanks the reviewer for this comment. For clusteroluminescence, it is a typical feature that the electron overlap or delocalization is absent in the occupied orbital but in the unoccupied orbital (see *JACS Au* **2021**, *1*, 1805-1814; *J. Am. Chem. Soc.* **2021**, *143*, 9565-9574; *Chem. Soc. Rev.* **2021**, *50*, 12616-12655). It is also the reason why its electron delocalization cannot be observed from absorption spectra but appears in the excitation spectra. LUMO is occupied by excitons after photoexcitation. This overlap can obviously increase delocalization and narrow the energy gap, resulting in redshifted emission with a longer wavelength than the intrinsic conjugation level of molecular structure. Hence, the electron delocalization in the LUMO can provide the necessary information to explain such an unconventional luminescence.

Based on the same calculation method and isovalue for orbitals, the electron overlaps of typical dimers of *EE-OBBO* and *EE-DMODA* could be observed by the naked eye (Fig. 6). *EE-OBBO* shows a greater number of overlaps (four for Dimer 5 and two for Dimer 6) than *EE-DMODA* (two for Dimer 1 and two for Dimer 2). Besides, the electron overlapping area of *EE-OBBO* is also larger than *EE-DMODA*. Moreover, the intermolecular distances in *EE-OBBO* are shorter than *EE-DMODA* as discussed in the manuscript. All of these parameters support that *EE-OBBO* showed stronger intermolecular TSI than *EE-DMODA*.

17. *The reviewer commented that “Page 9: I am wondering why the authors did not use a diyne building block in their work? This should yield a soluble/processable linear polymer.”*

Response: Thanks the reviewer for this consideration. We have tried to use a diyne building block to synthesize linear polymers. Interfacial polymerization cannot be used to synthesize linear polymers because the polymer film cannot be formed on the interface. However, when the polymerization was conducted in solution, only a tiny quantity of oligomers with very low molecular weights could be obtained.

18. *The reviewer commented that “Page 10, line 4: "the water concentration would be infinite". No. Pure water has a concentration of approximately 1000/18.01 M or 55.5 M under STP.”*

Response: Thanks the reviewer for this comment. We have revised the statement

as “the water concentration is about 55.5 M while the TMP concentration is 10^{-3} M, so r in equation 2 is extremely large, and $(p_A)_C$ approaches zero.” on page 11 of the revised manuscript.

19. *The reviewer asked that “Page 13: How heavy was the "drop ball"?”*

Response: Thanks the reviewer for this question. Its weight is about 80 mg, and the weight of our soft mechanical arm is about 20 mg in Fig. 10c. We have added their weights in the caption of Fig. 10 that “*Images of a soft mechanical arm (~ 20 mg) lifting a ball (~ 80 mg) upon exposure to DCM vapor and releasing it at 50 °C.*”.

20. *The reviewer commented that “ESI: Table 1 requires R1, WR2 and GOF values for the X-ray data presented. Otherwise, it is difficult for others to determine the quality of the X-ray refinement procedures.”*

Response: Thanks the reviewer for this comment. We have provided them in Supplementary Table 2 in the revised Supplementary Information.

21. *The reviewer commented that “ESI: How were the samples for the solid PL measurements prepared? Drop-cast from solution? etc..?”*

Response: Thanks the reviewer for this comment. The samples prepared through recrystallization were used for solid PL measurements. Two pieces of quartz were utilized to hold the sample in the instrument, as shown in Fig. R1.

Fig. R1 The crystalline or polycrystalline sample utilized for the solid PL measurements with two pieces of quartz as the holder.

22. *The reviewer commented that “ESI: Rf values should be given for all species purified by column chromatography.”*

Response: Thanks the reviewer for this comment. We have given them in the revised Supplementary Information.

23. *The reviewer commented that “Fig S8 and S32: I am wondering why the authors did not use TD-DFT to probe the excitations leading to emission? Use of TD-DFT is standard in the community these days.”*

Response: Thanks the reviewer for this comment. We have used TD-DFT to obtain the frontier molecular orbitals of *EE*-DMODA, *EZ*-DMODA, and *EE*-OBBO based on the optimized excited-state geometries as the reviewer suggested in Supplementary Figs. 8 and 37 in the revised Supplementary Information.

Supplementary Figure 8. Frontier molecular orbitals of (a) *EE*-DMODA and (b) *EZ*-DMODA based on the optimized excited-state geometries.

Supplementary Figure 37. Frontier molecular orbitals of *EE*-OBBO based on the optimized excited-state geometry.

24. *The reviewer commented that “ESI: It is odd that “56 mg” of alkyne was used in each reaction? Was this on purpose or perhaps a copy/paste error?”*

Response: Thanks the reviewer for this comment. “56 mg” is the weight of the organic catalyst DABCO.

Response to the comments and suggestions of Reviewer 2

The reviewer commented that “This manuscript describes the formation of clusteroluminescent compounds (CLgens) and polymers through a reaction of water molecules and alkynes in the presence of DABCO as an organic catalyst. Careful screening of the reaction using different types of water molecules such as heavy oxygen water (H_2^{18}O) and heavy water (D_2O) can suggest water involved reaction for the CLgen synthesis and the resulting molecular packing structures in their crystalline/solid-state control the through-space interactions (TSI). The small molecular CLgens’ synthesis discussion is clear and well written. In this study, the CLgens’ synthesis was applied at a water/organic solvent interface, and in the case of TMP as an alkyne reactant, the resulting product was 3D networking polymers. The new phenomenon of interfacial polymerization-induced emission (IPIE) sounds attractive.

Overall, although **this type of reaction is brand-new and attractive using water as a reactant to design nonaromatic fluorescent and vapor-responsive materials**, I judge this study needs further investigation in particular the polymers produced at the water/organic solvent interface. Other minor suggestions are listed below.”

We sincerely thank the reviewer for the strong support of the main findings in this study and constructive suggestions that helped us to further the quality.

1. *The reviewer commented that “For the structure analysis, IR and ^{13}C CP/MAS were utilized. How about further analysis by means of XPS, elemental analysis, XRD, TGA, and DSC? ”*

Response: Thanks the reviewer for this comment. The elemental analysis, XRD, TGA, and DSC test results and related descriptions have been added on page 11 in the revised manuscript and Supplementary Information. We are so sorry that we could not provide the XPS data, because the XPS in our university could not work. It has some serious problems and repairing it needs a long time. We believed that other measurements could already confirm the structures of PTMP.

The elemental analysis was performed on a ThermoFisher Scientific FlashSmart CHNS Elemental analyzer. Calcd for PTMP unit cell ($\text{C}_{30}\text{H}_{34}\text{O}_{15}$): C, 56.73%; H, 5.36%; Found: C, 56.40%; H, 6.25%.

The PXRD was collected on an X’per Pro (PANalytical) instrument at 25 °C (scan range: 3-30 °). The obtained PTMP is amorphous according to its PXRD pattern (Supplementary Figure 48 as shown below).

Supplementary Figure 48. PXRD pattern of PTMP.

The TGA was carried out on a SHIMADZU DTG-60AH analyzer under a nitrogen atmosphere at a heating rate of 10 K/min. The 5% weight loss temperature (T_d) is 268 °C for PTMP under nitrogen, indicating that it is thermally stable (Supplementary Figure 49 as shown below).

Supplementary Figure 49. TGA thermograms recorded under nitrogen at a heating rate of 10 K/min.

The DSC was carried out on a Mettler Toledo DSC3 analyzer under a nitrogen atmosphere at a 10 K/min heating rate. The glass transition temperature (T_g) is 82 °C (Supplementary Figure 50 as shown below).

Supplementary Figure 50. DSC thermograms of PTMP recorded under a nitrogen atmosphere during the first heating cycle at a scan rate of 10 K/min.

2. The reviewer commented that “There are discussions about crosslinking density of the polymeric films which depends on the polymerization time, however, no quantitative investigations such as porosity, surface area, density, etc, were carried out. Even AFM analysis, how are AFM images of 5 min and 20 min polymerization conditions? It may be possible to discuss more strictly the change of roughness evaluated from AFM. Is it really increasing the crosslinking (chemically) density or just aggregating the polymer and forming thicker films? In order to know this point, this reviewer recommends evaluating the polymer films by mechanical testing, such as a tensile test or bending test. Are they comparable to the 3D networking conventional polymers or elastomers? ”

Response: Thanks the reviewer for this comment. The reviewer provided many constructive suggestions for quantitative cross-linking density investigations.

The porosity and surface area results of PTMP are shown in Figs. R2-R4. Their porous properties are poor, and surface areas are low due to the flexible structures of PTMP.

Fig. R2 N₂ adsorption and desorption isotherms (77 K) of PTMP (5 min).

Fig. R3 N₂ adsorption and desorption isotherms (77 K) of PTMP (20 min).

Fig. R4 N₂ adsorption and desorption isotherms (77 K) of PTMP (60 min).

For the density test, because it needs too many samples, we are so sorry that we could not do this test.

For AFM analysis, we tested AFM images of 5 min and 20 min polymerization conditions and did roughness analysis as the reviewer suggested. Root-mean-square roughness (R_q) and average roughness (R_a) are given by Nanoscope Analysis 1.8 (Supplementary Figs. 60, 61 as shown below). When the polymerization time was prolonged, the R_q and R_a values decreased both on the water and DCM sides. It is also the evidence that tighter structures were formed and crosslinking density enhanced when the polymerization time was prolonged. We have added the roughness analysis on page 13 in the revised manuscript and Supplementary Information.

Supplementary Figure 60. Atomic force microscope of PTMP at different polymerization times on the water side (scale bar: 3 μ m).

Supplementary Figure 61. Atomic force microscope of PTMP at different polymerization times on the DCM side (scale bar: 3 μm).

For mechanical testing, the stress-strain curve for PTMP film (60 min) was obtained via TA Q800 dynamic thermal mechanical analyzer (DMA) (Supplementary Fig. 51). Its mechanical properties are comparable to reported crosslinked polymers obtained via the interfacial polymerizations (*Angew. Chem. Int. Ed.* **2022**, e202117390). We have added the related descriptions on page 11 of the revised manuscript. The PTMP films at 5 min and 20 min polymerization times could not be tested because of many defects in their films.

Supplementary Figure 51. Stress-strain curve for PTMP film (60 min) measured by tensile testing (1 N/min, room temperature, with the break point indicated by \times).

3. *The reviewer commented that “In the discussion about the actuator properties, I wonder about the responding speed and mechanical forces. Without such quantitative values, we are hard to imagine the performance and ability of the actuators.”*

Response: Thanks the reviewer for this comment. For the bending-recovering behavior of PTMP placed in an ethanol vapor or air atmosphere, it took about 20 s for bending and 15 s for recovering as shown in Supplementary Fig. 62. For the bending-recovering behavior of PTMP placed in a DCM vapor at 50 $^{\circ}\text{C}$, it took about 2 s for bending and 60 s for recovering as shown in Supplementary Fig. 63. The mechanical properties were given in Supplementary Fig. 51 as shown above. We have

added these values of responding speed in the revised manuscript.

Supplementary Figure 62. (a) Bending, and (b) recovering behavior of PTMP placed in an ethanol vapor or air atmosphere.

Supplementary Figure 63. (a) Bending, and (b) recovering behavior of PTMP placed in a DCM vapor or at 50 °C.

4. *The reviewer commented that “In the time-resolved PL decay analysis, Figs. 34-38 and Figs. 44-46, a curve of the instrument response function should be added in each figure. The amplitude-weighted lifetime components and fitting factor should be indicated as an accuracy explanation.”*

Response: Thanks the reviewer for this comment. We have given them in Supplementary Figs. 41-47 and 57-59 in the revised Supplementary Information.

5. *The reviewer pointed out that ““First” at the beginning of the Results is a typo.”*

Response: Thanks the reviewer for pointing out that. We have revised it in the revised manuscript.

6. *The reviewer commented that “It would be great if the authors study in-situ IPIE detection. How the emission change under growing the polymer film at the interface occur is interesting to know.”*

Response: Thanks the reviewer for this suggestion. Actually, we tried to check the in-situ IPIE phenomenon during the experiments. However, the polymeric film at the interface always showed very weak emissions due to the quenching effect of the solvent. Thus, it is challenging to detect the emission signal and study the in-situ IPIE phenomenon in this work at present.

Reviewers' Comments:

Reviewer #1:

Remarks to the Author:

The revised manuscript has satisfied all of my pre-existing questions, thus, I am happy to recommend publication of this nice study in Nature Comms. as is.

Reviewer #2:

Remarks to the Author:

The revised manuscript adequately responded to most of the reviewers' suggestions for revision and provided additional scientifically credible data. Therefore, I think the quality has reached enough to merit acceptance of the manuscript. However, here I suggest a few minor points to be revised further before its formal acceptance.

- 1) When measuring a solid sample's absorption and emission spectroscopy to be more reliable, I recommend that BaSO₄ powder be used as a matrix, and the solid be mixed in the matrix.
- 2) In Supplementary Table 1, the unit of the amplitude-weighted lifetime, τ , should be "ns" instead of "nm".
- 3) Since T_g appears at 82 °C, the generated cross-linked polymer is in a glassy state (amorphous) at room temperature. Thus, the result should be described with the result of PXR. In the current version, they were described separately. In addition, the heat capacity value for the glassy transition should be provided.

Point-by-Point Response to Reviewers' Comments

MS ID.: NCOMMS-23-01434A

MS Title: Facile conversion of water to functional molecules and cross-linked polymeric films with efficient clusteroluminescence

Response to the comments and suggestions of Reviewer 1

The reviewer commented that “The revised manuscript has satisfied all of my pre-existing questions, thus, I am happy to recommend publication of this nice study in Nature Comms. as is.”

We sincerely thank the reviewer for the strong support of the main findings in this study.

Response to the comments and suggestions of Reviewer 2

The reviewer commented that “The revised manuscript adequately responded to most of the reviewers' suggestions for revision and provided additional scientifically credible data. Therefore, I think the quality has reached enough to merit acceptance of the manuscript. However, here I suggest a few minor points to be revised further before its formal acceptance.”

We sincerely thank the reviewer for the strong support of the main findings in this study and constructive suggestions that helped us to further the quality.

1. *The reviewer commented that “When measuring a solid sample's absorption and emission spectroscopy to be more reliable, I recommend that BaSO₄ powder be used as a matrix, and the solid be mixed in the matrix.”*

Response: Thanks the reviewer for this comment. When our samples are mixed in BaSO₄ matrix, their cluster states and clusteroluminescence properties might be changed. Thus, we measure the sample's absorption and emission spectra in pristine crystalline or polycrystalline forms.

2. *The reviewer commented that “In Supplementary Table 1, the unit of the amplitude-weighted lifetime, τ , should be “ns” instead of “nm”.”*

Response: Thanks the reviewer for this comment. We have revised it.

3. *The reviewer commented that “Since T_g appears at 82 °C, the generated cross-linked polymer is in a glassy state (amorphous) at room temperature. Thus, the result should be described with the result of PXRD. In the current version, they*

were described separately. In addition, the heat capacity value for the glassy transition should be provided.”

Response: Thanks the reviewer for this comment. We described the DSC and PXRD results together as the reviewer suggested in the revised manuscript and provided the specific heat capacity difference (ΔC_p) in Supplementary Figure 49.